# Intrinsic Explainability of Multimodal Learning for Crop Yield Prediction

## Abstract

Multimodal learning enables various machine learning tasks to benefit from diverse data sources, effectively mimicking the interplay of different factors in real life events. While the heterogeneous nature of these modalities may necessitate the design of complex architectures, their interpretability is often overlooked. In this study, we leverage the intrinsic explainability of Transformer-based models to explain multimodal learning frameworks. We utilize the self-attention mechanism alongside model-specific feature attribution techniques, comparing these against post-hoc methods. Our detailed analysis focuses on the challenging task of crop yield prediction, exploiting the characteristics of the modalities and the data to aggregate local explanations at multiple levels. Our findings indicate that Transformers significantly outperform other architectures in yield prediction, making them well-suited for further intrinsic interpretability analysis. Among the modalities, satellite data emerged as the most influential but requires deeper layers for effective feature extraction due to its complex structure. Additionally, we observed that the Attention Rollout method is more robust than Generic Attention, aligns more closely with Shapley-based attributions and shows reduced sensitivity to minor input variations.

## 1 Introduction

Real-world events result from the interplay of multiple factors, with the combination of different information sources often needed to explain observed outcomes. This has led to the growing interest in multimodal learning within the Machine Learning (ML) community. This approach can leverage diverse sources of information to capture complex relationships and improve model performance across a wide range of tasks (Manzoor et al., 2023). In fact, models fusing data from different modalities outperform their uni-modal counterparts both intuitively and provably (Huang et al., 2021).

Despite its success, most work in the multimodal learning literature primarily focuses on designing complex architectures and optimizing performance, with limited emphasis on the interpretability of these models (Rahate et al., 2022). Given the often opaque nature of multimodal architectures, understanding how different modalities contribute to model predictions is crucial, particularly in high-stakes domains where decision-making relies on trust and transparency (Joshi et al., 2021).

In this context, intrinsic interpretability methods, which provide explanations directly tied to the model's internal components, offer a promising alternative to traditional post-hoc model-agnostic approaches that treat the model as a block box (Rudin, 2019). Intrinsic explanations are inherently more faithful and less prone to errors introduced by surrogate models (Ribeiro et al., 2016; Lundberg & Lee, 2017; Molnar, 2020). The need for intrinsic interpretability is especially relevant in Remote Sensing (RS) applications, where multiple data modalities—such as satellite imagery, climate and weather data, and topographical maps—are commonly used to predict complex environmental and agricultural phenomena (Mena et al., 2024a; Li et al., 2022; Günther et al., 2024; Rußwurm & Körner, 2020). Enhancing interpretability in these contexts can facilitate better understanding of how different factors (e.g., spectral bands, temporal dynamics) influence predictions, ultimately supporting more informed and actionable insights for practitioners. Accordingly, our work explores transparent, intrinsically interpretable multimodal networks for RS applications.

Section 2 provides an overview of prior work on explainability in multimodal learning networks, with a particular focus on leveraging self-attention mechanisms for explainability in RS applications. Section 3 outlines our modeling and explainability methodologies. In the results section, we describe the dataset in subsection 4.1, followed by the presentation of modeling outcomes in subsection 4.2. Subsequently, we address model interpretability by first analyzing the learned representations in subsection 4.3, followed by an investigation of temporal attributions and their association with specific weather events in 4.4, and the introduction of a modality importance estimation technique in 4.5. Finally, we conclude with a summary of the findings in Section 5.

## 2 RELATED WORK

Explainability in multimodal learning networks has gained increasing attention as these models allow the combination of diverse data types, yet the difficulty of this task results in complex architectures and threatens the interpretability of their decision-making processes (Joshi et al., 2021). Feature attribution techniques, such as SHAP (Lundberg & Lee, 2017) and Integrated Gradients (Sundararajan et al., 2017), are model agnostic explanation techniques, and can thus easily be applied to multimodal networks. Recently, graph-based explainability methods have been proposed to model inter-modality dependencies more comprehensively (Ghosh et al., 2019; Gaur et al., 2021). Other methods leverage attention mechanisms to highlight the importance of different modalities and their interactions, yet such applications often only visualize the attention weights of certain input samples, which provides very limited insights into the more general understanding of the model (Ghosal et al., 2018; Tsai et al., 2019).

The RS field is particularly rich in modalities, making the explainability of multimodal learning in this context crucial for sensitive applications including disaster management, environmental monitoring, agriculture (Günther et al., 2024). One particularly challenging RS agricultural application is crop yield prediction. Predicting crop yield is a particularly challenging task due to the involvement of multiple factors. Due to scarcity of labeled data, this problem is often addressed at the field or regional level, leaving the sub-field level relatively underexplored (Leukel et al., 2023; Murugananatham et al., 2022; Nevavuori et al., 2019). The application of multimodal learning at both levels can be classified into studies that employ either an early-fusion approach (Cai et al., 2019; Gavahi et al., 2021; Wang et al., 2020; Cao et al., 2021) or those that apply a modality-specific encoding of the data before applying an intermediate or late fusion of the learned representations (Pathak et al., 2023; Ma et al., 2023; Yang et al., 2019; Maimaitijiang et al., 2020; Jeong et al., 2022; Mena et al., 2024b).

Taking a closer look at the use of self-attention mechanisms to leverage their inherent interpretability in RS, researchers have explored this approach for several tasks, including crop classification (Khan et al., 2024; Xu et al., 2021; Rußwurm & Körner, 2020; Garnot et al., 2020; Obadic et al., 2022), land cover classification (Kim et al., 2022; Méger et al., 2022), water quality monitoring (Pyo et al., 2021), and target detection (Zhou et al., 2019). However, the analysis of self-attention mechanisms for eXplainable AI (XAI) in these studies is often limited, with little focus on in-depth interpretability. In the context of yield prediction, while many studies have utilized attention-based models to enhance task accuracy (Inderka et al., 2024; Krishnan et al., 2024; Qiao et al., 2023; Lin et al., 2023; Junankar et al., 2023), we could identify only one study which has explicitly focused on explaining such models. Tian et al. (2021) used an attention-based long short-term memory (ALSTM) model, which combines a LSTM network with an attention layer, to predict winter wheat yield at the county level in central China. However, this study does not leverage the attention mechanism for inherent explainability and instead relies on post-hoc methods.

Our work demonstrates how the attention mechanism, particularly in Transformer-based models, can be leveraged to enhance the intrinsic interpretability of multimodal networks. We conduct our analysis on the yield prediction task, contributing in the following four aspects: **1. model interpretability**: we leverage the inherent interpretability of the attention mechanism to explain yield predictions. **2. post-hoc vs. intrinsic**: we also apply model-agnostic explanation methods and compare against the intrinsic explanations. **3. multimodal learning**: we incorporate four modalities with rich variables, processed individually before applying an intermediate fusion of learned representations. **4 sub-field yield modeling**: we utilize extensive yield records from Argentina for three different crops, making predictions at the sub-field with a 10m resolution.

## 3 METHODOLOGY

### 3.1 MODELING

This section outlines the models used for crop yield prediction based on pixel-wise processing of the spatially aligned modalities. We test various neural network architectures for encoding individual modality information and fusing the learned representations.

**Modality Encoder**  Depending on the modality's nature, i.e., static or temporal, we use different neural network architectures to encode its representation. For static modalities, such as the terrain elevations and soil properties, we use multilayer perceptrons (MLPs). For temporal modalities, such as satellite and weather data, we test four different types of architectures: long short-term memory (LSTM) (Hochreiter & Schmidhuber, 1997), ALSTM (Tian et al., 2021), 1-Dimensional convolutional neural networks (1D-CNNs) (Zheng et al., 2016; Pelletier et al., 2019), and Transformers (Vaswani et al., 2017). Each of these modality encoders is expected to produce a representation denoted as $h \in \mathbb{R}^d$.

**Feature Fusion**  Given the heterogeneous nature of the input modalities usually used in RS and other fields, intermediate-level fusion is well-suited for our study, as opposed to input-level fusion (Liang et al., 2024; Mena et al., 2024a). We test three simple yet effective feature fusion methods. First, a simple *concatenation* along a new dimension can be applied to the learned representations. Second, a *scaled dot-product attention (SDPA)* mechanism can be employed to compute attention-based weights, which are then used to perform a weighted sum over the representations (Vaswani et al., 2017). Finally, a *cross-attention* fusion approach can be implemented, where a Transformer block integrates the modality representations considering them as sequence tokens. The fusion operation is followed by a linear regression layer to predict the yield. The training process and the hyperparameter tuning for each architecture are detailed in Appendix A.3.

### 3.2 EXPLAINABILITY

An important contribution of our study is the interpretation of multimodal networks. In the following, we describe the various tools used to explain the yield prediction model, with particular emphasis on intrinsic interpretability in Transformer-based architectures.

**Attention layers dynamic**  To better understand the roles and dynamics of the intermediate layers, we use linear classifier probes (Alain & Bengio, 2016). In practice, linear probes consist of linear regressors that take as input the latent features learned by an intermediate layer of the trained model and learns to predict the corresponding yield value, as predicted by the model. High accuracy of this regressor suggests a linear separability of the features at the examined layer. By comparing the accuracy of linear probes across successive layers, we can verify whether the learned features gradually become more separable, thus facilitating the final prediction.

**Self-Attention mechanism**  Since the introduction of attention mechanisms in the literature, many have seen the opportunity to use the weights for explaining neural networks (Vaswani et al., 2017; Rußwurm & Körner, 2020; Xu et al., 2021). Indeed, the attention weights link the input to the subsequent layers of the network, allowing the model to focus on relevant parts of the input for performing a specific task, and this link is used to interpret the model reasoning behind individual predictions.

**Attention Rollout**  In a multi-head multi-layer Transformer block, each sample generates multiple attention weight matrices. Direct analysis of each matrix can be time-consuming and might not easily reveal the inner workings of the model. Additionally, as we progress to higher layers within the model, the identifiability of individual time steps decreases, resulting in increasingly mixed information. Consequently, direct probing of attention weight matrices for explainability becomes unreliable. Therefore, to trace the information propagated from the input layer to the final embeddings of each Transformer block, we employ Attention Rollout (AR) (Abnar & Zuidema, 2020). This method treats attention weights as proportion factors and iteratively multiplies the attention

weight matrices of the multiple attention layers. The resulting matrix encodes the attention distributions of the entire Transformer block and can thus serve as a reliable basis for explanation. In our analysis, we specifically focus on the attention weights corresponding to the regression token.

**Generic Attention** Another approach that leverages the internal workings of the Transformer model and facilitates its interpretation is Generic Attention (GA) (Chefer et al., 2021). Unlike AR, which only uses the attention weight matrices, GA propagates information backward from the final output through the last Transformer layer and subsequently through all preceding layers using gradients. As with AR, our analysis will focus on the resulting weights that attend to the regression token.

**Post-hoc feature attribution** Shapley values (Shapley, 1953), a concept derived from cooperative game theory, are commonly applied in the field of XAI as a model-agnostic method. In contrast to AR and GA, Shapley values are estimated using only the input samples and treating the model as a black box. This is achieved by masking certain features, passing the modified sample through the model, and measuring the change in prediction. To mitigate the high computational cost of computing exact Shapley values, we employ their approximation technique Shapley Value Sampling (SVS) (Strumbelj & Kononenko, 2010). SVS has demonstrated superior robustness in terms of sensitivity and fidelity compared to other attribution methods on similar yield prediction tasks (Yeh et al., 2019; Najjar et al., 2023).

## 4 EXPERIMENTS AND RESULTS

### 4.1 DATA

To predict crop yield, target values are collected using combine harvesters across Argentina for three crops: corn, soybean, and wheat. These three datasets provide geo-referenced yield values in tons per hectare (t/ha) at the subfield level and spans multiple years (2017–2023). For modeling purposes, the yield maps are rasterized to a 10-meter spatial resolution, to match the corresponding satellite images from the Sentinel-2 (S2) mission. Our analyses will mainly focus on the corn dataset, which includes 21 farms, 147 fields, and a total of more than one million data points. More details describing the remaining datasets and the yield preprocessing steps are provided in Appendix B.1. In addition to the time series of satellite data, the input modalities include weather, soil, and digital elevation map (DEM). Further details on yield data preprocessing and the input modalities are provided in Appendix B.

### 4.2 MODEL EVALUATION

**Architecture type** To assess the performance of the models described in Section 3 and Appendix A, we first evaluate their coefficient of determination ($R^2$) scores on the validation set to select the best-performing models. In Table 1, we present the scores of the best model from each architecture type on the test set, including mean absolute error (MAE).

Table 1: Comparison of model performance evaluated on the test set.

| Model | # Parameters | R² | MAE |
|---|---|---|---|
| 1D-CNN | 6,437,505 | 0.28 | 2.24 |
| LSTM | 54,977 | 0.41 | 2.00 |
| ALSTM | 38,017 | 0.41 | 2.00 |
| Transformer | 109,345 | **0.46** | **1.90** |

We observe that the Transformer model achieves the highest accuracy, followed closely by the AL-STM and LSTM models at the subfield level (i.e.pixel level). However, when comparing field-level averages of target and predicted values, the difference in performance between these three architectures becomes more pronounced at the field level, where the Transformer model demonstrates a clear advantage, and the attention mechanism also improves the performance of the recurrent network. We attach field-level scores in Appendix C. In contrast, the best 1D-CNN-based model fails to achieve a comparable performance. Finally, the clear superior performance of the Transformer-based architecture, combined with its inherently interpretable attention mechanism, strongly supports the opinion that improving interpretability in ML does not necessarily require compromising model performance (Rudin, 2019).

**Transformer configuration**    To further investigate the behavior of different configurations of the Transformer-based model, we compare its performance when changing the number of layers or heads, or using a different fusion block from the best-performing architecture - which has *four layers and a single-head for both temporal modalities, and uses a concatenation-based approach* for the fusion. We notice that these changes have a relatively minor impact on overall performance. More details are provided in Appendix D. An important implication of this observation is that the selection of model architecture can prioritize simplicity and ease of interpretability over marginal gains in evaluation metrics. Specifically, using single-head Transformer blocks and a concatenation-based fusion facilitates the model interpretation, contrary to averaging the results across multiple heads (Abnar & Zuidema, 2020; Chefer et al., 2021).

We also analyzed the similarity of the representations learned for each modality across various configurations using the Singular Vector Canonical Correlation Analysis (SVCCA) technique. Contrary to our expectations, the results show that retraining the same model with a different random initialization seed or varying the Transformer hyperparameters leads to significantly different learned representations. Although we will not explore this aspect further, we can mention here that our analysis of the variance captured by the top singular vectors suggests that the weather data encoding can be represented with a much smaller vector compared to the satellite modality. We have included the detailed results of this analysis in Appendix D.

**Qualitative results**    To visually compare the performance of the best model from each architecture type, we selected a field from the validation set, referred to as Field-A, where all models achieve a moderate to good accuracy, and visualize its target, prediction, and error maps. In Figure 1, the top row displays the target yield values (a) alongside the predicted values from the best-performing 1D-CNN (b), LSTM (c), and Transformer (d) models. The second row shows the corresponding error maps for each model. For the 1D-CNN model, we notice that the model fails to predict varying yield values, failing to accurately capture the variance observed in the target, especially in the bottom half of the field, where the yield is under-estimated. This issue is highlighted in the 1D-CNN error map, where large differences between the predicted and actual values are shown in red. In contrast, the LSTM and Transformer models demonstrate better performance, with both models more closely matching the target yield variance. However, discrepancies remain in certain high-yield zones. Notably, the range of values in the error map for the Transformer model is smaller compared to that of the LSTM model, indicating that the Transformer is better at minimizing prediction errors across the field.

We conducted the same analysis on a field where the Transformer model demonstrated poor performance, referred to as Field-B, and observed a similar relative behavior among the different architectures, with the Transformer still outperforming the others. This suggests that even under less favorable conditions, the Transformer model retains a comparative advantage in performance. The corresponding prediction and error maps are provided in Appendix D. *Due to its consistent superior performance, the subsequent analyses will primarily focus on the Transformer architecture.*

### 4.3 PROBING LEARNED REPRESENTATIONS

In this section, we evaluate the information content of intermediate model representations using linear probing. Next, we analyze the attention weight matrices learned by the model, evaluating their similarity for pixels within the same field and examining how these weights are distributed across the different layers of the Transformer encoders. This analysis focuses on the best-performing Transformer-based model.

**Linear Probing**    We investigate the linear separability of the intermediate layers of the best-performing Transformer model. To facilitate this analysis, we randomly select 100,000 samples, representing approximately 10% of the data, using 90% of these samples to train linear probes and the remaining 10% for testing. For each layer, we compute its output given the selected samples as inputs, flatten these latent representations, and then use them to train a linear model to predict the model's final yield prediction. The Root mean square error (RMSE) scores on the test set are presented in Figure 2.

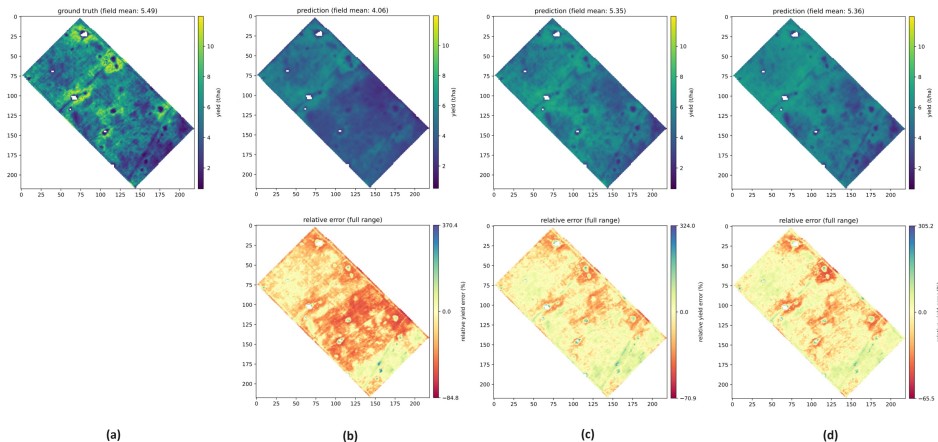

Figure 1: Ground-truth (a) and predicted (b-d) yield values from the best performing model of each architecture (1D-CNN, LSTM and Transformer, respectively) on Field-A.

We observe that the intermediate representations learned for the satellite data demonstrate the highest linear correlation to the predicted values, followed closely by the weather data. In contrast, soil and DEM data show a significantly lower linear correlation. Given the static nature of these two modalities, they are processed using shallow MLPs, and they also have low spatial resolution, which contributes to their limited potential to predict the yield. When comparing the temporal modalities, i.e. satellite and weather data, the results indicate that the linear separability of weather data remains nearly constant throughout the Transformer layers, whereas a significant increase is observed across the satellite encoder layers. This trend can be attributed to the higher complexity of the satellite

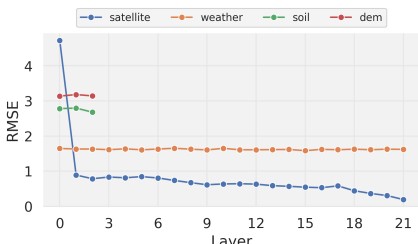

Figure 2: RMSE test scores of the linear probes attached to the modality encoders.

time series, which has the highest spatial resolution and comprises 12 spectral bands, in contrast to the four weather properties used. This observation aligns with the SVCCA findings mentioned in the previous subsection, where only a few principal singular vectors were sufficient to capture the weather data's variance.

**Attention weights: In-field distribution** Considering that yield variations are expected to be minimal and growth conditions are similar for pixels within the same field, we quantify the similarity of attention weights at the field level to later aggregate the attention-based explanations at this level. This analysis is conducted through the following steps: First, 200 pixels are randomly selected from each field. Then, the (i) cosine similarity of the attention weights and the (ii) difference in predicted yield are calculated for each pair of pixels, separately in each field. For the last layer we only compare the attention weights attending to the regression token. Finally, scatter plots are generated, where the similarity values are plotted per field and colored according to the corresponding absolute error.

An example in Figure 3 illustrates the results from each layer of the satellite Transformer encoder from 20 random fields. For the first three layers, the distance between the flattened full attention weight matrices is compared, whereas for the final layer, only the weights attending to the regression token are considered. We notice a pronounced similarity in the first layer, but it progressively diminishes in the deeper layers of the block. Additionally, no correlation is found between the absolute prediction error and the distance between the attention weights of the compared pixel pairs. This suggests that similar predictions are not necessarily associated with a similar distribution of attention across different time steps, even for pixels within the same field.

We also conducted the same analysis to compare the AR and GA results. As shown in Figure 4, a significant correlation is noted between the AR attributions at the field level, in contrast to the larger differences observed with GA. This suggests a higher robustness of AR compared to GA, as the high similarities observed in layers 1 and 2, in Figure 3, should not be entirely outweighed by the decreasing similarities in subsequent layers. Additionally, a desirable property of attribution methods is low sensitivity, meaning that minor variations in input feature values should not lead to significant changes in the attributions (Yeh et al., 2019). Since pixels from the same fields typically experience similar environmental conditions, their input values are expected to be comparable, and consequently, their attributions should exhibit consistency as well. The inclusion of gradients in the computation of GA could contribute to its high sensitivity. For the weather encoder, we observe perfect similarity across all evaluated fields, irrespective of the method used. This is attributed to the low spatial resolution of weather data, often leading to identical input weather values for all pixels within the same field. The corresponding plots are provided in Appendix E.1.

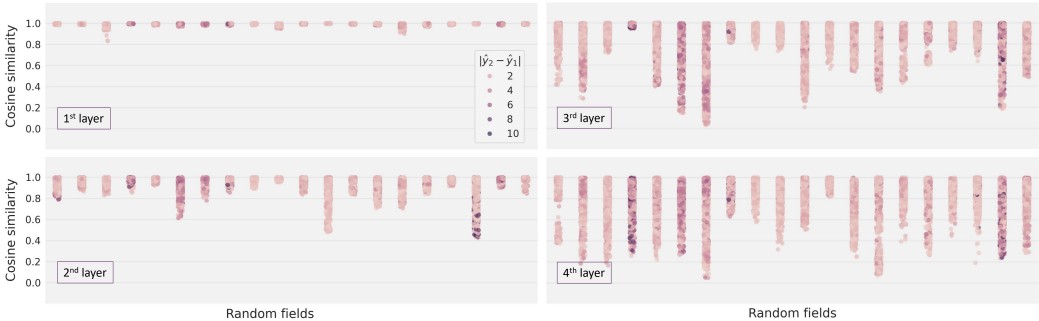

Figure 3: Cosine similarity of the attention weights from the satellite transformer encoder of multiple pairs of pixels in a consistent set of 20 random fields, and the corresponding difference in prediction.

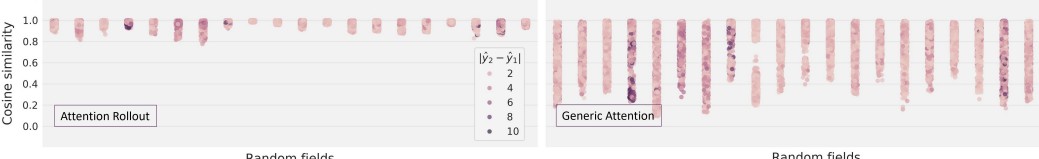

Figure 4: Cosine similarity of the AR and GA of the satellite transformer encoder of multiple pairs of pixels in a consistent set of 20 random fields, and the corresponding difference in prediction.

**Attention weights: Layer-wise distribution** After assessing the similarity of the attention weights across different pixels, we now study their temporal distribution across different layers. We use the raw, bi-dimensional, attention matrix and sum the attention weights allocated to each time step, which allows us to determine the total attention each time step receives from all other steps. This process is repeated for each layer to understand how attention is distributed throughout the network. Exceptionally for the final layer, we take a different approach: we directly evaluate the attention weights that lead to the regression token, since all other time steps are disregarded in subsequent processing by the model. This approach provides insight into which time steps are prioritized by the model as it makes its final prediction. Figure 5 presents these results for the temporal modalities, with Field-A shown in the top row and Field-B in the bottom row. More fields are displayed in Appendix E.

In the case of the satellite time series, as depicted in Figure 5.a, we observe that the attention weights from the first layer (represented in blue) are distributed smoothly across the entire time series. This indicates that the first layer does not distinctly discriminate between different time steps, implying a more generalized initial processing. In contrast, the subsequent three layers show a marked shift, each assigning higher attention weights to specific time steps, indicating a focus on different growth periods. These difference across layers were also observed in similar previous studies (Xu et al., 2021). Moreover, the varying patterns of attention distribution across different fields suggest that each layer might be capturing unique temporal dynamics relevant to the conditions of each field.

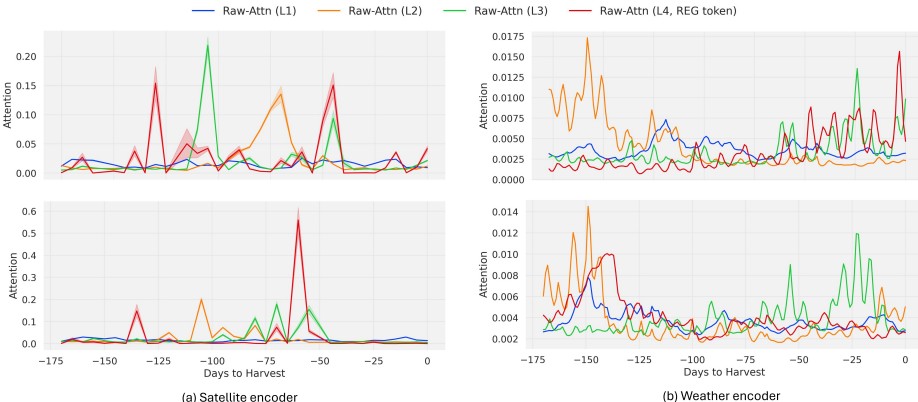

(a) Satellite encoder    (b) Weather encoder

Figure 5: Total attention weights attending at each time step for the first 3 attention layers, and the regression token weights in the final layer. The results are averaged across 200 randomly selected pixels from Field-A, at the top, and Field-B, at the bottom, and are displayed for the satellite (a) and weather (b) Transformer encoders. The light buffer regions represent the 95% confidence interval around the average value.

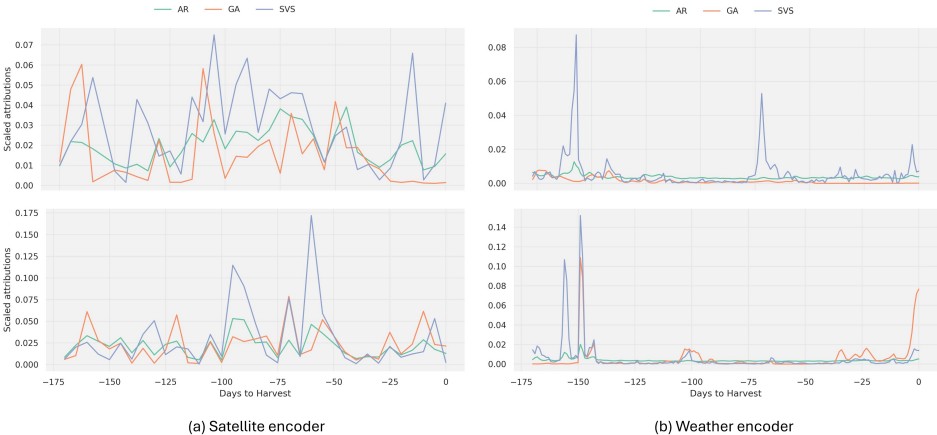

(a) Satellite encoder    (b) Weather encoder

Figure 6: Field-level average attributions of the satellite (a) and weather (b) modalities. Fields A and B are displayed in the top and bottom rows, respectively.

For the weather encoder, the attention distribution results shown in Figure 5.b reveal that all layers exhibit a discriminative behavior across different time steps. Unlike the satellite encoder, each layer consistently emphasizes certain time periods, suggesting a continuous refinement of temporal focus throughout the layers. This could be attributed to the longer sequences in the weather data, which necessitate the use of multiple layers to attend to different growth periods, ensuring comprehensive temporal coverage and detailed focus throughout the growth cycle.

These findings highlight the differential use of attention mechanisms across modalities and how different layers of the Transformer model specialize in capturing various temporal aspects of the data, providing insights into how the model interprets and prioritizes different parts of the time series for yield prediction.

### 4.4 TEMPORAL ATTRIBUTIONS

**Attribution methods comparison**    We analyze here the temporal attributions provided by the AR and GA methods, and compare them against the SVS scores. Due to the high computational cost associated with the SVS method, we limited the number of pixels sampled per field to 32 pixels. Figure 6 displays the average attributions for Field-A and Field-B. A visual assessment of the results

for the satellite encoder reveals patterns that are consistent across all three methods. In contrast, within the weather encoder, the SVS method appears to play a more discriminative role compared to the AR and GA methods. This indicates that SVS may be more sensitive to temporal variations in weather data. Results for additional fields are provided in Appendix F. To quantitatively assess the similarity between the different attribution methods, we calculate the cosine similarity between each pair of methods based on the field-level averaged attributions, and display the results in Figure 7. When comparing modalities, we observe consistently higher similarity scores for satellite data compared to weather data, indicating that the methods align more closely when estimating temporal attributions for the satellite signal. When comparing methods, SVS and GA methods exhibit the lowest similarity, suggesting that AR is the most effective in approximating the behavior of model-agnostic methods. This aligns with the high similarity observed between SVS and AR for both modalities.

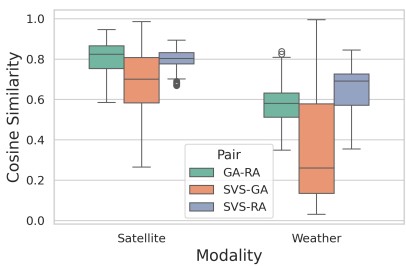

Figure 7: Distribution of field-level cosine similarities between every pair of the compared attribution methods: GA, AR and SVS.

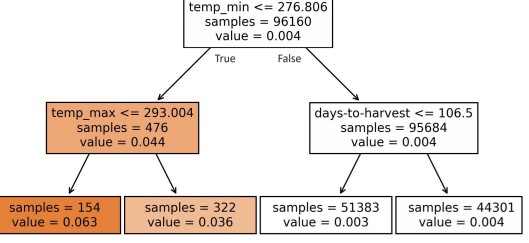

Figure 8: Decision Tree with two levels. The results shown are on the train set of 3 fields from the same farm, from 2021, predicting the AR temporal attributions of the weather Transformer encoder. The color of each box is used as a scale for the predicted attribution values.

**Weather Events** To investigate the possible impact of special weather events on their attributions, we train a decision tree model to predict the attribution of each time step based on its weather properties: minimum, average, and maximum daily temperatures, as well as total precipitation. We additionally include the number of days before harvest as a predictive feature, allowing the model to contextualize each weather event within the growth cycle of the crop. We train a separate decision tree for each set of fields belonging to the same farm and the same year, as described in Appendix G. We experiment with decision tree depths of two and three, to ensure the learned models remain interpretable. We report the results of the models that predict the AR attributions. Figure 8 shows the results for a farm with three fields from 2021 where the accuracy was particularly high and thus reliable for interpretation, reaching 89% in the training set and 90% in the test set. In the tree, we observe that the right branch predominantly covers time steps with attribution values of 0.003 or 0.004. These low-importance events are characterized by a minimal daily temperature above 276.8 and constitute 99.5% of the training samples. Conversely, the darkest leaf in the tree, representing only 0.16% of the dataset (154 samples), shows a notably high attribution score of 0.063. These high-importance events are associated with both minimal and maximal temperatures below 276.8 and 293 K, respectively. A slight increase in the tree depth allowed the tree to achieve better performance across multiple farms while maintaining interpretability. In Appendix G, we extract similar insights using a tree with three levels trained on a different farm. These analysis and findings are generally useful in identifying weather events that significantly influence the decisions made by the Transformer model, highlighting the critical role that specific temperature conditions play during particular days of the crop growth period.

### 4.5 MODALITY IMPORTANCE

**Weighted modality activations** Since the best performing Transformer model uses a concatenation-based fusion block followed by a linear layer, we propose to exploit its structure to infer modality impact score. We can rewrite the final prediction $\hat{y}_i$ of sample $i$ as the weighted combination of the modality activations $\mathbf{z}_i = \text{concat}(\mathbf{z}_i^m)$, with $m \in \{\text{satellite, weather, soil, dem}\}$ and infer modality relevance scores $\mathcal{R}_i^m$:

$$\hat{y}_i = \mathbf{w}.\mathbf{z}_i + b = \sum_m \mathbf{w}^m.\mathbf{z}_i^m + b = \sum_m \hat{y}_i^m + b, \quad \mathcal{R}_i^m = |\frac{\hat{y}_i^m}{\hat{y} - b}|$$

where $\mathbf{w} = \text{concat}\left(\mathbf{w}^{sa}, \mathbf{w}^w, \mathbf{w}^{so}, \mathbf{w}^d\right)$ and $b$ are the weights vector and bias of the final regression layer, respectively. This approach can be viewed as an alternative to Class Activation Mapping (CAM) and Gradient-weighted CAM (Grad-CAM) methods (Zhou et al., 2016; Selvaraju et al., 2017), which are widely used for explaining classification tasks in computer vision. However, while CAM and Grad-CAM are specifically designed for convolutional networks operating on a single modality, our method is applicable to any multimodal regression task utilizing a concatenation fusion mechanism and a MLP as a regression head. Furthermore, it can be extended to various differentiable fusion strategies and regression heads through gradient-based techniques. We compare this method to Shapley-derived modality scores, since SVS can estimate the contribution of all individual input features to the model's prediction. We describe the corresponding aggregation process in Appendix H.

**Modality impact** In Figure 9, we compare both methods and present the modality scores for 50 corn fields. The weighted modality activations indicates that soil features have the highest impact on the prediction, accounting for an average of 37.8% across all fields, followed by satellite data at 28% and weather data at 24%. Terrain elevation features contribute the least, with an average impact below 10%. In contrast, Shapley values indicate a different distribution of relative importance, with satellite data contributing the predominant share at 72.3% on average, followed by weather at 24.6%. We attach in Appendix H the results of the same comparison for wheat and soybean

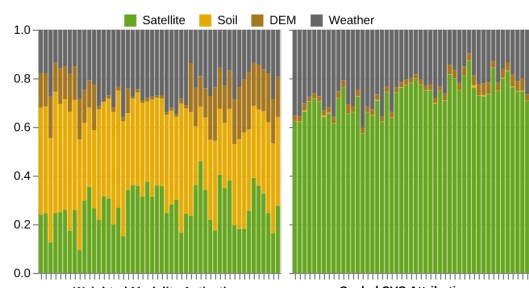

Figure 9: Comparing the modality scores for the same random set of 50 corn fields.

fields, in which the differences among both compared techniques are consistent. This difference can be particularly attributed to the computational process. The weighted averages rely only on the regression head to infer modality scores, while the SVS method uses the entire model. Shapley values stand out due to their ability to capture feature interactions by employing principles from game theory, considering multiple feature subsets and their contributions to the model before inferring feature attributions. In contrast, the strength of weighted activations lies in their inherent connection to the model's architecture, which makes their importance estimations more faithful to the model's behavior (Rudin, 2019). Overall, evaluating the correctness of these methods is challenging, as the modality impact scores do not necessarily reflect the agronomic significance of each modality, where established field knowledge could have been leveraged as a reference. Instead, these scores indicate how the model uses each modality, which depends on its learning scheme.

## 5 CONCLUSION

We attempt in our work to highlight the potential of leveraging intrinsic interpretability within transformer-based models to enhance understanding in multimodal learning frameworks. We examined the learned representations for each modality, inferred temporal attributions using both model-specific and model-agnostic approaches, and proposed an intrinsic method to derive modality importance scores. Our analysis, conducted on the challenging task of yield prediction, underscored the varying information complexity across input modalities and its influence on the learned representations and attention weights. The comparative evaluation of the temporal attribution methods revealed distinct patterns, indicating the need for further evaluations. Our proposed approach for inferring modality importance offers deeper insights into how the model uses different data sources, the method can be extended to other fusion techniques, thereby enhancing transparency in more complex multimodal architectures. We hope our findings advance the state-of-the-art in interpretable multimodal learning, offering practical implications for deploying trustworthy models in critical, data-rich domains like environmental and agricultural monitoring.

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

# A MODELING

## A.1 MODALITY ENCODERS

To process multiple modalities, we test different architectures which first encode each modality individually before fusing the learned representations. Figure 10 depicts the different architecture types used. In the following subsections, we provide a concise overview of the modality encoders utilized in this study, along with the fusion techniques applied and the hyperparameters fine-tuning approach.

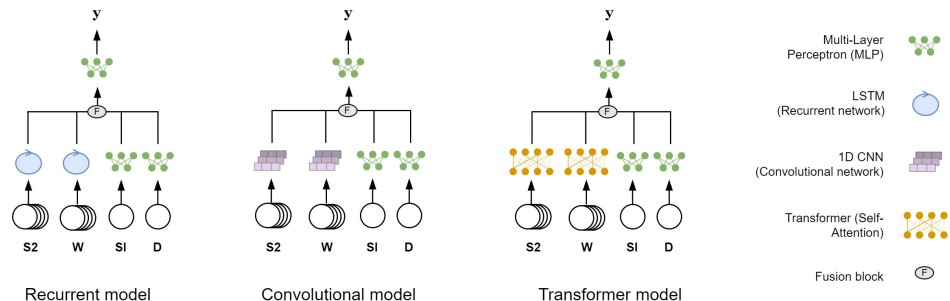

Figure 10: Multimodal architectures with intermediate fusion for yield prediction.

### A.1.1 MULTI-LAYER PERCEPTRON

MLPs are a type of artificial neural network where information flows in one direction, from the input layer through hidden layers to the output layer, without any loops or cycles. MLPs extract features by learning high-level representations through layers of neurons, each performing a weighted sum followed by a non-linear activation function. In our implementation, we use two fully connected layers: the first layer has a dimension of $4d$, and the second layer has a dimension of $d$, which returns the modality representation. Batch normalization and the ReLU activation function are applied after the first layer.

### A.1.2 RECURRENT NEURAL NETWORKS

Recurrent neural networks (RNNs) are inherently capable of handling temporal data. They process one time step at a time, learning to predict outputs and maintain a hidden state at each step. The hidden state is optimized to focus on important information while discarding irrelevant or redundant data. In our implementation of the RNN, we use a stack of two LSTM cells (Hochreiter & Schmidhuber, 1997) with a dropout rate of 0.3, followed by a linear layer to transform the LSTM output at the final time step to a dimension of $d$. Before applying the linear layer, we use batch normalization to improve training stability.

We also explore another RNN variant based on ALSTM Tian et al. (2021), which aggregates outputs from all time steps using a weighted combination, rather than relying solely on the final time step. The weights are computed using a form of scaled dot-product attention (Vaswani et al., 2017).

### A.1.3 CONVOLUTIONAL NEURAL NETWORKS

A 1D-CNN is primarily used for analyzing sequential data by applying convolutional filters across one-dimensional input, such as time series or signals. In 1D-CNNs, information flows directly from the input layer to the output layer without loops or recurrences. Unlike RNNs, which process one time step at a time, 1D-CNNs use convolutional filters to capture patterns or features along the temporal dimension of the input. Our implementation follows the feature extraction approach used in TempCNN (Pelletier et al., 2019), with the modification of using a linear layer at the end instead of a SoftMax layer to produce a modality representation of dimension $d$.

### A.1.4 TRANSFORMER

Transformers are highly effective for modeling temporal data due to their ability to use self-attention mechanisms (Bahdanau et al., 2016; Vaswani et al., 2017) to capture long-range dependencies within the input sequence. Unlike RNNs and 1D-CNNs, which process data sequentially or locally, Transformers attend to all time steps simultaneously, allowing them to more effectively capture complex temporal patterns. In our implementation, we utilize a Transformer-based model (Vaswani et al., 2017) for temporal data encoding. The input features are first passed through a linear embedding layer, which transforms each time step into a token of size $d$, while a learnable regression token similar to *class token* in (Devlin, 2018; Dosovitskiy et al., 2021) is added to interact with all time steps. Before adding the regression token and feeding the data to the Transformer layers, positional encoding is applied based on the date of the time step. We use two calendar years, covering the crop season, and for each time step, we calculate the number of days from the beginning of the first year to determine its index. This positional encoding follows the approach of (Vaswani et al., 2017), except we use the index calculated as described. The transformed input is then processed through multiple layers of Transformer encoders, each consisting of multi-head self-attention (MHA) and position-wise feed-forward networks. In each Transformer layer, the input undergoes layer normalization before being processed through MHA. The output from the MHA layer is added back to the input via a residual connection, followed by a second layer normalization step. A position-wise feed-forward network is then applied, with its output also added through residual connections. This process is repeated across several Transformer layers, with the final modality representation derived from the output of the class token.

## A.2 INTERMEDIATE FUSION

### A.2.1 CONCATENATION

In concatenation fusion, feature vectors from each modality are concatenated along the feature dimension. If there are $m$ modalities, each with dimensionality $d$, the resulting fused feature representation will have a dimensionality of $m \cdot d$.

### A.2.2 SCALED-DOT PRODUCT ATTENTION

We apply scaled dot-product attention (Vaswani et al., 2017; Miranda et al., 2024), where the input representations from multiple modalities serve as both keys and values, and a learnable vector serves as the query. Each modality representation is treated as a token, and these tokens are stacked to form the keys and values for the scaled dot-product operation. Mathematically, this is expressed as follows, with the learnable query vector $\boldsymbol{q} \in \mathbb{R}^d$, and the stacked keys from the $m$ modalities represented by $K \in \mathbb{R}^{m \cdot d}$:

$$\text{SDP Attention Fusion}(\boldsymbol{q}, K) = \text{softmax}\left(\frac{\boldsymbol{q}K^T}{\sqrt{d}}\right) K \tag{1}$$

### A.2.3 CROSS-ATTENTION

In cross-attention fusion, we leverage a multi-layer, multi-head transformer encoder to fuse representations from multiple modalities. Each modality is represented as a token, and these tokens are stacked into a sequence and fed into the transformer. We introduce a learnable regression token

Table 2: Yield data description. We train different models for each country-crop pair.

| Country | Crop | Years | # Farms | # Fields | # Pixels |
|---|---|---|---|---|---|
| Argentina | corn | 2017-2023 | 21 | 147 | 1,003,133 |
| Argentina | soybean | 2017-2023 | 29 | 289 | 2,103,250 |
| Argentina | wheat | 2017-2022 | 13 | 61 | 497,651 |

that interacts with all modality tokens across transformer layers. This token aggregates information through attention, evolving into a fused representation that captures both modality-specific and cross-modal features, resulting in a richer, more expressive representation for downstream tasks.

### A.3 MODEL FINETUNING

The different model architectures incorporate multiple hyperparameters that can influence model performance. We experimented with various configurations of hidden sizes, numbers of attention heads and layers, and feature fusion techniques to optimize performance for the yield prediction task. For this purpose, the dataset was split into training, validation, and test sets, with the validation set used to select the best network configuration, and the test set used to evaluate and report the model's performance on unseen data.

The models were trained using mini-batch stochastic gradient descent with the Adam optimizer and decoupled weight decay (Loshchilov, 2017). We employed a learning rate scheduler that begins with a linear warm-up for 5 epochs, followed by cosine decay for 50 epochs (Loshchilov & Hutter, 2022). Early stopping was implemented to stop training when the validation loss did not decrease for 10 consecutive epochs.

## B DATA

### B.1 YIELD DATA

Yield maps derived from data collected by combine harvesters are used as ground truth. As the combine harvester traverses the field, it records equidistant data points at a high spatial resolution, with each point characterized by various features such as geographic coordinates, yield in t/ha, and yield moisture in percentage.

To harmonize the raw yield data, we employ a standardized preprocessing pipeline. This includes reprojecting the coordinate reference system, standardizing feature naming conventions, and removing erroneous entries related to position, timestamp, yield, moisture, and non-operational harvesters. Additionally, zero-yield points and agronomically unrealistic values are filtered out. Data points are further refined using statistical thresholds to ensure that yield values remain within three standard deviations.

The processed point vector data is subsequently rasterized into 10-meter resolution yield maps, aligned with the corresponding satellite imagery raster data. An overview of the utilized yield datasets is provided in Table 2.

### B.2 INPUT MODALITIES

We use 4 modalities to address the yield prediction task; Satellite data, from S2 mission, Weather data, DEM and soil properties. The satellite data contains 12 spectral bands (i.e., channels), while the weather data includes minimum, average, and maximum temperatures, as well as total precipitation. Soil and DEM modalities include 8 and 5 static properties, respectively. Although the spatial resolutions of all four modalities was aligned for pixel-wise yield prediction, the original temporal resolutions are maintained: satellite data follows an approximately 5-day revisit interval, while weather features are represented as daily averages. Tables 3 and 4 summarize the features in each input modality, along with its spatial and temporal resolutions. For static features, only the spatial resolution is provided. In Table 4, twi, cec, cfvo, phh2o and soc stand for topographic wetness index, cation exchange capacity, volumetric fraction of coarse fragments, soil pH and soil organic carbon, respectively.

Table 3: Characteristics of satellite and weather features, with corresponding temporal (Tp.Res.) and spatial (Sp.Res.) resolutions.

| Modality | Dynamic features | Source | Sp.Res. | Tp.Res. |
|---|---|---|---|---|
| Satellite | B01 - Coastal Aerosol | S2 | 60 m | 5 days |
| | B02 - Blue | S2 | 10 m | 5 days |
| | B03 - Green | S2 | 10 m | 5 days |
| | B04 - Red | S2 | 10 m | 5 days |
| | B05 - Red Edge 1 | S2 | 20 m | 5 days |
| | B06 - Red Edge 2 | S2 | 20 m | 5 days |
| | B07 - Red Edge 3 | S2 | 20 m | 5 days |
| | B08 - NIR | S2 | 10 m | 5 days |
| | B8A - Narrow NIR | S2 | 20 m | 5 days |
| | B09 - Water vapour | S2 | 60 m | 5 days |
| | B11 - SWIR 1 | S2 | 20 m | 5 days |
| | B12 - SWIR 2 | S2 | 20 m | 5 days |
| Weather | Max temperature | ERA5 | 30 km | Daily |
| | Mean temperature | ERA5 | 30 km | Daily |
| | Min temperature | ERA5 | 30 km | Daily |
| | Total precipitation | ERA5 | 30 km | Daily |

Table 4: Characteristics of soil and terrain elevation features, with corresponding spatial resolutions (Sp.Res.).

| Modality | Static features | Source | Sp.Res. |
|---|---|---|---|
| DEM | Elevation | SRTM | 30 m |
| | Slope | SRTM | 30 m |
| | Curvature | SRTM | 30 m |
| | TWI | SRTM | 30 m |
| | Aspect | SRTM | 30 m |
| Soil | CEC | SoilGrids | 250 m |
| | CFVO | SoilGrids | 250 m |
| | Nitrogen | SoilGrids | 250 m |
| | pHH2O | SoilGrids | 250 m |
| | Sand | SoilGrids | 250 m |
| | Silt | SoilGrids | 250 m |
| | SOC | SoilGrids | 250 m |
| | Clay | SoilGrids | 250 m |

### B.3 DATA SPLITTING

Since each sample represents a pixel from a field, we grouped samples by field to ensure that the model encounters unseen fields in the validation and test splits. To maintain a consistent data distribution, we stratified the splits by year, ensuring that each split contains data from all years.

## C MODEL EVALUATION

We evaluate the different multimodal networks on both field and subfield levels, and report the $R^2$, MAE and relative root mean square error (RRMSE) scores in Table 5.

We further illustrate the performance results by visualizing the target, prediction and error maps. Figure 11 depicts the results for Field-B, in which the Transformer model did not perform very well. However, a similar relative behavior is observed as compared to Field-A. The yield map (b) generated by the 1D-CNN model shows a limited ability to capture the yield variances present in the target map (a), while the LSTM and Transformer models, shown in maps (c) and (d), respectively, capture more of the yield variances seen in the target, as evidenced by their corresponding error maps. Despite the overall lower performance in Field-B, the range of error values for the Trans-

Table 5: Comparison of model performance evaluated on the subfield-level (i.e. pixel level) and the field-level on the test set.

| Model | # Parameters | Subfield-Level | | | Field-Level | | |
|---|---|---|---|---|---|---|---|
| | | R² | MAE | RRMSE | R² | MAE | RRMSE |
| 1D-CNN | 6,437,505 | 0.28 | 2.24 | 0.36 | 0.47 | 1.49 | 0.22 |
| LSTM | 54,977 | 0.41 | 2.00 | **0.29** | 0.52 | 1.40 | 0.19 |
| ALSTM | 38,017 | 0.41 | 2.00 | 0.31 | 0.67 | 1.20 | 0.17 |
| Transformer | 109,345 | **0.46** | **1.90** | **0.29** | **0.70** | **0.98** | **0.16** |

former model remains narrower than those of the 1D-CNN and LSTM models, indicating that even in less favorable conditions, the Transformer model maintains a better performance.

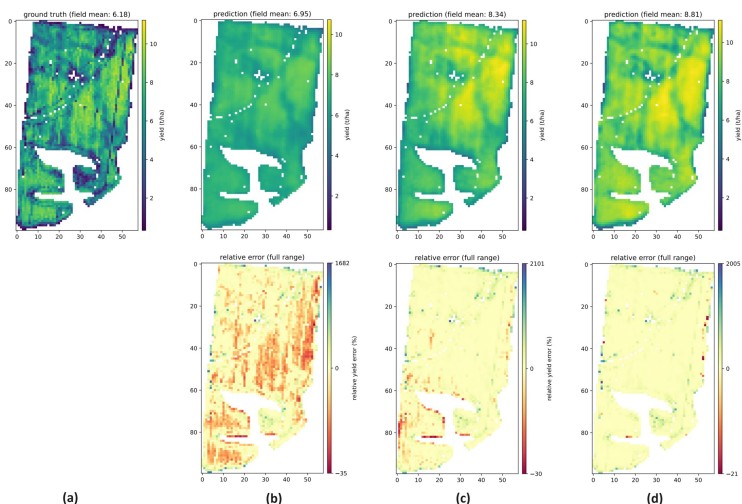

Figure 11: Ground-truth (a) and predicted (b-d) yield values from the best performing model of each architecture (1D-CNN, LSTM and Transformer, respectively) on Field-B.

## D  TRANSFORMERS CONFIGURATIONS

**Comparing performance**  To investigate the behavior of different configurations of the Transformer-based model, we report the evaluation metrics on the validation and test sets across various model setups, as shown in Table 6. We begin with the best-performing architecture, listed in the first row, which employs a configuration of 4 layers and a single-head Transformer encoder for both temporal modalities (i.e., satellite and weather time series). This model uses a concatenation-based approach to fuse the modality-specific representations. From this baseline configuration, we either increase the number of heads or layers, or alter the fusion approach to include a simple scaled-dot-product operation or a full Transformer block (with parameters similar to those of the modality encoders).

Our observations indicate that the performance differences between the compared models are not significant, particularly when examining the test set results. Interestingly, some variants outperform the best model configuration on the test set, despite the best model performing optimally on the validation set. This suggests that changes in model parameters, such as the number of heads or layers, and variations in the fusion approach, have a relatively minor impact on overall performance. All model configurations achieve scores within a narrow range, indicating that the Transformer-based models are robust across different configurations and that their performance does not heavily depend on these specific architectural choices.

**Comparing representations through SVCCA**  SVCCA is a general method proposed by Raghu et al. (2017) for efficiently comparing the learned representations between different neural network

Table 6: Comparison of Transformer models performance evaluated on the subfield-level (i.e. pixel level) and the field-level.

| Model | Val-Subfield-level | | Val-Field-level | | Test-Subfield-level | | Test-Field-level | |
|---|---|---|---|---|---|---|---|---|
| | R² | MAE | R² | MAE | R² | MAE | R² | MAE |
| 1H-4L-concat | 0,77 | 1,34 | 0,92 | 0,52 | 0,46 | 1,90 | 0,70 | 0,98 |
| 1H-4L-cross-attn | 0,71 | 1,54 | 0,8 | 0,87 | 0,56 | 1,70 | 0,79 | 0,95 |
| 1H-4L-sdp-attn | 0,73 | 1,49 | 0,86 | 0,83 | 0,50 | 1,83 | 0,68 | 1,15 |
| 1H-6L-concat | 0,73 | 1,45 | 0,88 | 0,77 | 0,56 | 1,70 | 0,76 | 0,95 |
| 2H-4L-concat | 0,72 | 1,51 | 0,81 | 0,88 | 0,52 | 1,77 | 0,79 | 0,89 |
| 4H-4L-concat | 0,73 | 1,46 | 0,85 | 0,79 | 0,51 | 1,75 | 0,69 | 0,99 |

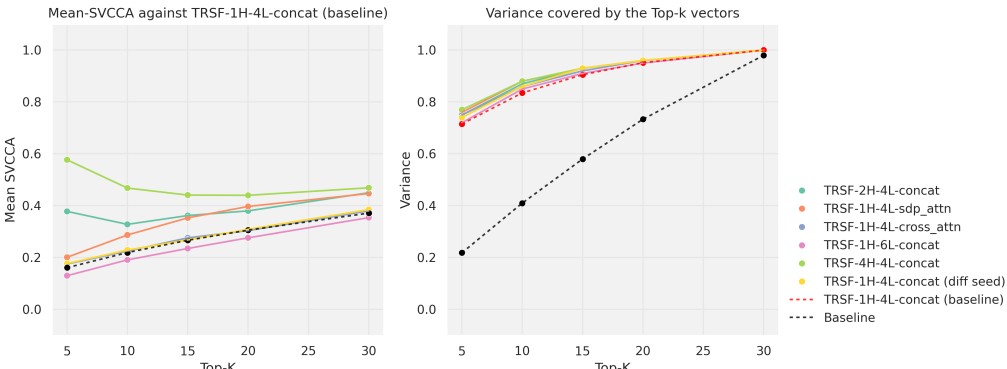

Figure 12: SVCCA results comparing learned representations for the **satellite** modality by the Transformer models against the best performing instance. On the left are the correlations between the top-k main vectors from each final layer of the corresponding compared models, and on the right are the variances captures by these main vectors in each model individually. *TRSF* refers to Transformer-based model, *H* and *L* respectively indicate the number of heads and layers in the Transformer encoders, while *concat*, *sdp_attn* and *cross_attn* refer to different fusion approach, as described in Appendix A.2.

layers and architectures, in a way that is both invariant to affine transform and fast to compute. We use SVCCA to compare the embeddings learned for each modality across different networks, focusing on the satellite and weather modalities.

SVCCA mainly consists of two steps. First, singular vectors for each model are obtained by applying singular value decomposition (SVD). Subsequently, canonical correlation analysis (CCA) is applied to compute the correlation coefficients between the aligned singular vectors (Hardoon et al., 2004). These vectors are ordered in descending order based on the variance they capture, and the correlations of the top $k$ vectors are averaged to obtain mean-SVCCA values for different values of $k \in \{1, 2, \ldots, d\}$, as illustrated in Figures 12 and 13.

In our study, all Transformer encoders used map each modality to a vector of 32 elements. To retrieve the learned representations of the satellite and weather modalities, we randomly select 160 samples from the input data, which is five times the vector length (as recommended by the authors of SVCCA), and process them through each model pair being compared. We evaluate the best-performing model against its variants, which differ by fusion head type, the number of layers in the Transformer encoders, or the number of Transformer heads. As a baseline, we generate a random representation of 32 elements for each sample, following a standard normal distribution. This random representation serves as a reference against which we compare the learned representations of the best-performing model. Additionally, we train a second instance of the best-performing architecture with a different random initialization of the weights and compare the representations obtained from both models.

In Figure 12, the mean-SVCCA values for $k \in 5, 10, 15, 20, 30$ are displayed on the left side. Notably, three experiments show similar or inferior results compared to the baseline curve, indicating

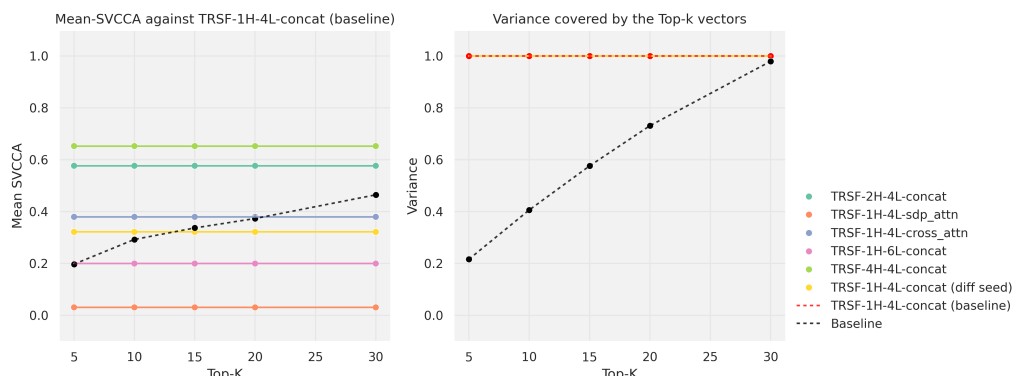

Figure 13: SVCCA results comparing learned representations for **weather** by the Transformer models against the best performing instance. On the left are the correlations between the top-k main vectors from each final layer of the corresponding compared models, and on the right are the variances captures by these main vectors in each model individually.

that the representations learned by our best model and these three experiments are weakly correlated. Specifically, the yellow line illustrates how a different initialization of the model can result in the learning of significantly different representations for the same modality, the purple line indicates that altering the fusion head from simple concatenation to a Transformer block significantly changes the prior representations learned for the satellite data, while the pink curve reflects even lower correlation when the satellite Transformer encoder is modified to include two additional layers. In contrast, a more positive correlation is observed when the number of heads is increased to two or four, as illustrated by the blue and green curves, respectively. This finding aligns with the work of Voita et al. (2019), which suggests that multi-head configurations can be unnecessary, as some heads may not learn additional relevant information. In our results, the model with a single head outperformed those with multiple heads.

Figure 13 illustrates the results of the same analysis conducted on the weather data encoder. The relative correlation of the different architectures to the best-performing model is similar to the findings from the satellite data encoder. Additionally, the weather data exhibits an interesting and consistent behavior: the top five singular vectors are sufficient to capture the complete variance of the 32-dimensional representation, in all examined architectures. This observation suggests that the information encoded in the weather data possesses considerably lower complexity compared to that of the satellite data.

## E    ATTENTION WEIGHTS DISTRIBUTION

### E.1    IN-FIELD DISTRIBUTION

We examine the similarity of attention weights of the weather Transformer encoder, following the same procedure described in Section **??**. The results are shown in Figure 14 for the raw attention matrices, and Figure 15 for the AR and GA attributions. As previously noted, the low spatial resolution of weather data often results in identical weather feature values across all pixels within the same field, which explains the perfect similarity scores observed in Figures 14 and 15.

### E.2    LAYER-WISE DISTRIBUTION

Figure 16 displays the comparison of attention weights distribution across different layers of the Transformer encoder of satellite and weather modalities, for random corn fields.

## F    TEMPORAL ATTRIBUTIONS

In figure 17 we compare the temporal attribution methods for random corn fields.

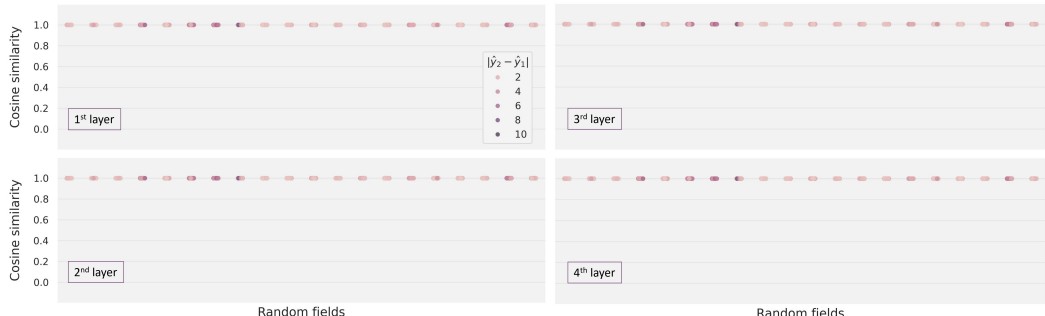

Figure 14: Cosine similarity of the attention weights from the weather transformer encoder of multiple pairs of pixels in a consistent set of 20 random fields, and the corresponding prediction absolute error. For the first three layers, we evaluate the cosine similarity between the full attention weight matrices, while for the last layer we only compare the attention weights attending to the regression token.

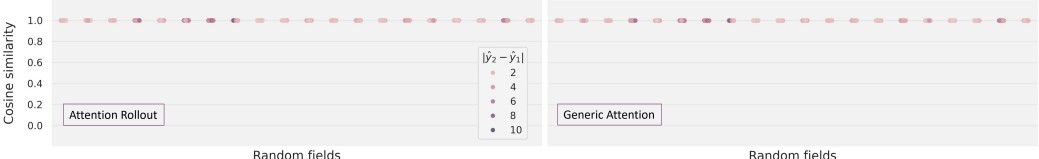

Figure 15: Cosine similarity of the AR and GA of the weather transformer encoder of multiple pairs of pixels in a consistent set of 20 random fields, and the corresponding prediction absolute error.

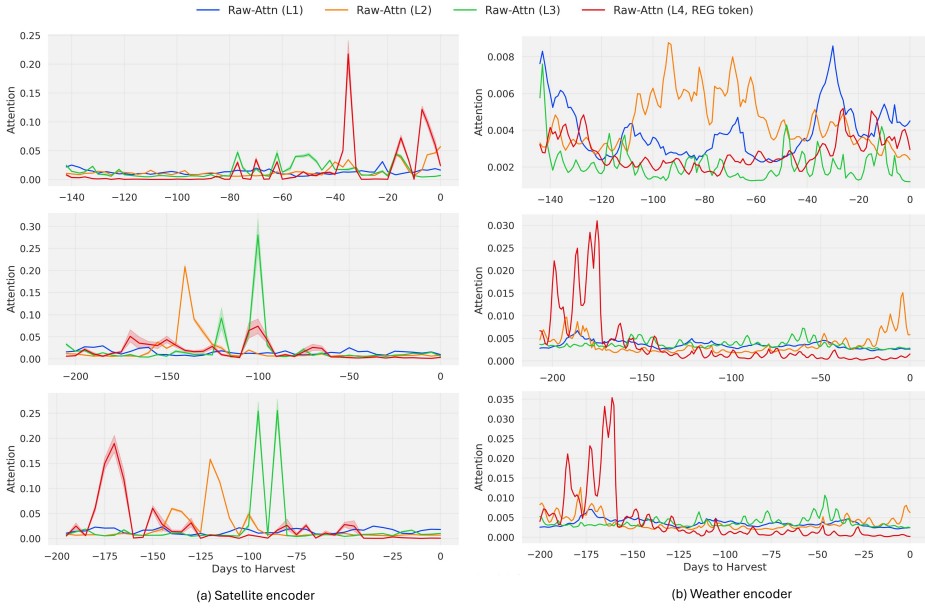

Figure 16: Total attention weights attending at each time step for the first three attention layers, and the regression token weights in the final layer. The results are averaged across 32 randomly selected pixels from three random fields, and are displayed for the satellite (a) and weather (b) Transformer encoders. The light buffer regions represent the 95% confidence interval around the average value.

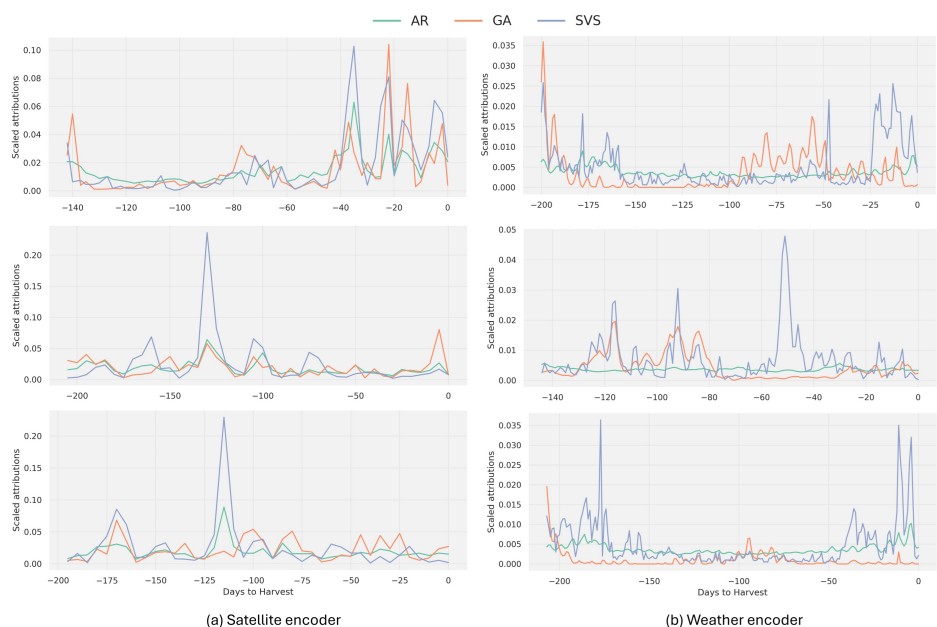

(a) Satellite encoder          (b) Weather encoder

Figure 17: Field-level average attributions of the satellite (a) and weather (b) modalities, for three random fields.

## G WEATHER EVENTS

We examine the correlation between particular weather events and their attributions by training decision trees. For each set of fields belonging to the same farm and the same year, a separate tree is trained and a dataset is created including corresponding weather properties and the number of days before harvest as a predictive feature. Specifically, we randomly sample 200 pixels from each field, merge the associated weather time series, shuffle the instances, and then partition the datasets into 80% for training and 20% for testing. The AR attribution for each time step is used as the target variable.

Figure 18 illustrates the weather events decision tree for a farm of three fields from the year 2023. For this farm, the tree model achieved an accuracy of 83% on the training set and 84% on the test set on the task of predicting the AR temporal attributions.

We observe that the right branch of the tree covers a large portion of the training samples, greater than 90%, and indicates that all weather events occurring 19 days or more before the harvesting date have low importance, with attribution values not exceeding 0.006. This suggests that weather conditions far from the harvest date played a minimal role in influencing yield predictions made by the Transformer-model.

In contrast, the left branch identifies a specific subset of 942 events (0.9% of the samples) that were assigned high importance. Analyzing the rules leading to this leaf, we can conclude that during the 18 days before harvesting, days with maximum daily temperature between 287.46 and 287.92 K receive the highest attribution value of 0.01. This finding indicates that such weather events are highly influential in the Transformer model, suggesting a critical role that specific temperature conditions play in the days leading up to harvest.

## H MODALITY IMPORTANCE

**SVS-based modality importance** SVS results include the contribution of each individual input features. To infer the relative importance of different data modalities, we aggregate the Shapley values across features from each modality. Specifically, we first compute for each pixel the importance score of each input feature by taking the absolute values of the SVS scores, which are then summed

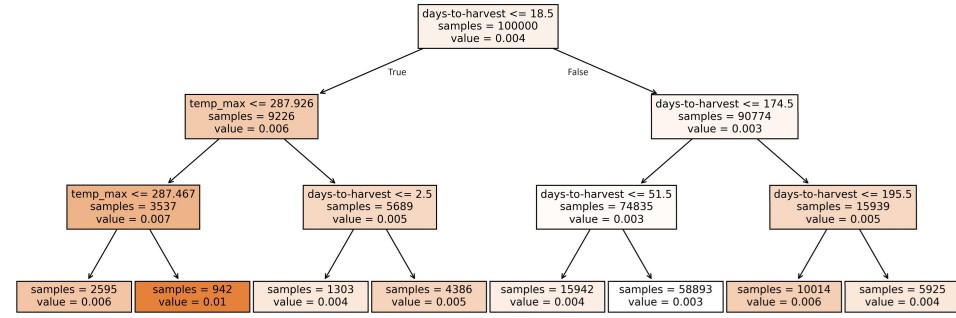

Figure 18: Decision Tree with three levels. The results shown are on the train set of 3 fields from the same farm, from 2021, predicting the rollout attention temporal attributions of the weather Transformer encoder.

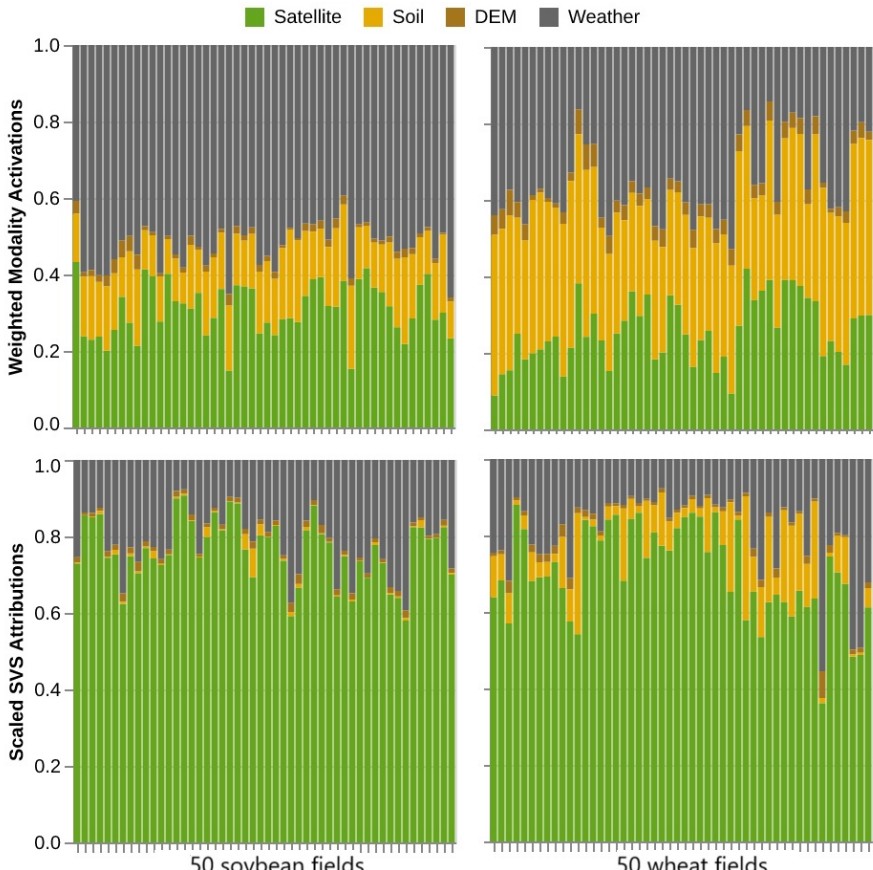

Figure 19: Comparing the modality importance using the weighted modality activations and the aggregated SVS scores for 50 soybean and wheat fields.

separately for each modality. To ensure comparability, we subsequently scale the modality scores so that they sum to one. This modality scoring process is repeated across a random selection of 32 pixels per field, using the same pixel samples as in Section 4.4. We then aggregate the scores per field by averaging the scores of each modality across the 32 samples.

**Additional results** Figure 19 compares the weighted modality activations and SVS scores for 50 fields from soybean and wheat crops. Similarly to corn fields, we observe that satellite data the most influential modality according to Shapley-based scores, and has much less impact according to the

weighted activations. This latter technique highlights the significant influence of weather conditions in soybean fields, and a comparable importance of soil in wheat fields.

