# OpenReview forum: "Intrinsic Explainability of Multimodal Learning for Crop Yield Prediction"
_ICLR.cc/2025/Conference — ICLR 2025 Conference Withdrawn Submission_

### Official Review · Reviewer_R7fa · 2024-10-28

**Soundness:** 3
**Presentation:** 3
**Contribution:** 2
**Rating:** 5
**Confidence:** 3

**Summary:**

The paper examines the intrinsic explainability of Transformer-based models for multimodal crop yield prediction, using satellite, weather, soil, and digital elevation map data. By focusing on explainability, the study addresses the challenge of understanding model decisions in agriculture. It uses self-attention layers to make predictions more interpretable and explores both intrinsic and post-hoc methods like Attention Rollout and Shapley values for feature attribution. The paper suggests that transformer models outperform alternatives like LSTM and CNN in yield prediction accuracy and interpretability, especially for satellite data, which was identified as the most influential modality.

**Strengths:**

The paper is well-written. The clarity in structure and flow makes complex concepts, such as the attention mechanism and the distinctions between interpretability methods, accessible.

The results indicate practical utility, with a focus on regions like Argentina and crops such as corn and soybean, making the findings potentially valuable for real-world applications in crop yield forecasting.

**Weaknesses:**

w1: The paper highlights discrepancies between the weighted modality activations and Shapley values for feature importance. For instance, soil features receive the highest weight with the regression head method (37.8%) but are less prominent in Shapley-based scores, which prioritize satellite data at 72.3% for corn yield predictions. This inconsistency can lead to ambiguity for practitioners who rely on model interpretation, as it’s unclear which modality consistently influences predictions .


w2: The proposed model shows variability in capturing yield variance between different fields. In Field-B, where the Transformer model struggled, it still outperformed other architectures, yet error maps reveal inconsistencies in how yield variance is captured compared to Field-A. This suggests the model may not generalize well across varying field conditions, especially in geographically diverse regions

**Questions:**

Q1: Given the varying significance of the satellite and weather data, could you provide more on how attention allocation informs actionable decisions for farmers or agronomists?


Q2: You use weather-based decision trees to identify high-attribution events. Could these temporal attributions be further refined to identify specific agricultural milestones (e.g., planting, harvesting)?

Q3: While this study is focused on crop yield, could the interpretability framework be adapted to other agricultural applications, such as disease detection or segmentation tasks?

---

> ### Author Response · Authors · 2024-11-19
> **Part 1/2**
>
> We thank the reviewer for their valuable feedback.
>
> **W1: inconsistencies between SVS and weighted modality activation results**
>
> In XAI evaluation studies, such inconsistencies are often highlighted when comparing multiple explanation methods. It is, in fact, more convenient for authors to sidestep these discrepancies by interpreting the model using a single method. However, how does this benefit the research community? Instead, we have included and compared results from different methods to emphasize the challenge of achieving consistent modality importance estimates: post-hoc methods do not align with intrinsic methods. This underscores a critical point: such modality importance results are not yet ready to be shared with practitioners in the farming field. Instead, we urge the XAI community to conduct further research to resolve this ambiguity and determine which methods provide more accurate and faithful feature attribution estimates, particularly in the context of multimodal learning, where such studies remain scarce.
>
> **W2: variance in Field-B is not accurately captured**
>
> We see your point, though we interpret it differently. In Field-B, the first and last maps in the first row actually exhibit similar variance: regions of high yield in the first map also appear as the lightest in color in the second map. The primary difference lies in the offset. As noted in the description above the map (for which we’ll increase the font size in the final revision), the ground truth has an average yield of 6.18 t/ha, while the Transformer predictions average 8.81 t/ha. This results in visually similar variances but with an offset in yield values and thus in the color shading.
>
> In fact, the model captures the variance in most fields, as we confirmed by double-checking our results. This is particularly impressive given that the model processes the data pixel-wise and never sees the satellite image of a field as a whole. Nevertheless, to address this point further, we will include the Bhattacharyya metric in the final revision. This metric compares distributions without being affected by offsets, and we will add the corresponding values to the figures.
>
> **Q1: how attention allocation informs actionable decisions for farmers or agronomists?**
>
> Typically, the goal of explaining the model is first to understand the model's reasoning, and second to verify it against established domain knowledge. For instance, if the model attributes high importance scores to weather events that agronomic literature identifies as influential on crop growth and yield, this alignment suggests that the model has captured meaningful patterns between the input data and the target. This increases our confidence in the model's ability to generalize to new data and make reliable predictions after deployment.
> However, when the model's reasoning diverges from expert knowledge, two possible interpretations arise: (i) the model may have learned spurious correlations, which could undermine its accuracy, particularly on unseen data; or (ii) the model may have identified patterns that are agronomically valid but not yet recognized by experts in the field. The latter scenario opens opportunities for scientific discovery derived from explainability results, though these findings require thorough interdisciplinary studies to verify and confirm their validity.
>
> In our work, the results demonstrate that the model focuses on different time steps in the satellite and weather data. This discrepancy is not contradictory but rather expected, as the model extracts different cues from each data source to predict the yield: satellite pixels provide a "visual" cue representing how the crop develops, while weather features offer yield factor-related information, such as the effects of temperature and precipitation on growth.
> In other words, the model leverages the satellite data to observe crop growth patterns and uses weather features to estimate the weather conditions influence the growing environment, and thus final yield, and the combines these complementary inputs to make its predictions. To support this explanation, we included in the rebuttal revision an analysis of specific soybean fields where phenological information was available, as detailed in the general comment.

---

> ### Author Response · Authors · 2024-11-19
> **Part 2/2**
>
> **Q2: can weather temporal attributions help identify specific agricultural milestones such as planting and harvesting?**
>
> In theory yes, this might be possible, but based on the satellite data. If time series inputs are not constrained to span from seeding to harvesting dates, the model could potentially learn to identify whether a field has been planted or not, relying on the visual cues provided by the satellite data. As a result, the temporal importance of satellite data is expected to increase starting with the first image captured after planting. This could occur a few days after seeding, depending on the satellite's temporal resolution (e.g., 5-day intervals). Thus, temporal importance from satellite data could help approximate the planting date, with a potential slight offset due to this resolution.
>
> However, the weather data provides a different type of information and would not serve as a reliable proxy for identifying planting or harvesting dates. Weather patterns prior to seeding, for example, can still influence the model's predictions, as these contribute to soil preparation and establish the climate conditions for the crop's early growth stages.
>
> While this type of analysis could provide valuable insights, it is beyond the scope of our current study, and because our dataset contains satellite and weather data strictly limited to the seeding-to-harvesting window, we can’t run these experiments yet.
>
> **Q3: could the interpretability framework be adapted to disease detection or segmentation tasks?**
>
> The framework proposed is applied in the paper on time series data for a regression task. Thus it can easily be applied to any multimodal network for a similar application.
>
> 1. **For classification tasks** using time series data, namely disease classification, AR can directly be applied, since it only need the attention weights in its analysis, while GA and SVS methods include the predicted output in their calculations, which implies that applying them to a classification task will provide separate feature attributions for each individual class prediction. Typically, practitioners limit this analysis to either the target class or the one with the highest predicted probability to reduce complexity.
>
> 2. **For segmentation tasks**, the model takes as input an image (or time series of images), while our framework is applicable to tabular time series (note that in our work we process the satellite images pixel-wise: each pixel is a sample, passed individually to the model). Processing images will require different types of architectures than our Transformer-based models, thus our framework cannot directly be applied, and will need significant modifications to be adapted to computer vision tasks. Additionally, segmentation tasks are in general challenging to interpret, because of the nature of their output: the model returns a map of values, and each value can be explained individually, but how to aggregate these explanations is challenging.
> Though our expertise with this task is limited, we can refer you to this [study](https://ieeexplore.ieee.org/document/10436675), which uses a gradient-based explanation method for predicting sea ice concentrations from satellite images. It may provide insights into handling explainability for segmentation problems.

---

### Official Review · Reviewer_6E8s · 2024-10-31

**Soundness:** 3
**Presentation:** 4
**Contribution:** 2
**Rating:** 6
**Confidence:** 3

**Summary:**

This paper assesses how varied interpretability and explainability techniques can be combined in a comprehensive approach to understand model-wise / layer-wise / time-wise / modality-wise behaviour of the learned model. In particular, it evaluates both representations and attention scores at the level of layers, compared to Attention Rollout and Generic Attention on each modality.

**Strengths:**

- This paper is well structured and understandable.
- It has thoroughly referred to related paper, both in establishing existing approaches and problem significance, and to corroborate the results of the study.
- Each of the interpretability evaluation provides visualisations, which can be leveraged to suggest either particularities of the model architecture or qualities of the data that might influence the model.
- The dataset used is very large, and authors make sure that the analysis benefit from geographical units within the dataset (fields, farm, subfields).
- A systematic analysis uses multiple techniques to analyze each model component individually or globally.

**Weaknesses:**

- Only one dataset is used, with no variation on modalities or indicators within each modalities. Even within the dataset, the authors do not truly enter a discussion on concept drift between the different fields, despite observing variability. Time-wise drift is not evaluated, despite years going from 2017 to 2023. How generalizable the approach is to other regions and future years is unknown.
- Expert understanding is missing from the evaluation, hence several of the explanations that the authors provide about data influence on the model cannot be verified. Sections after lines 371, 416, 460, and 501 lack this expert confirmation on data behaviour. Additionally, assertion line 419 could have been verified experimentally by computing attention distributions on 5 days averages, or 5 days min and max, instead of daily series. This would require fitting a new model on the modified data.
- No statistical significance, despite seemingly a lot of samples. If not samples (pixels), subfields could have been used to conduct statistical tests over distribution means.
- Despite actionability being mentionned in introduction, no such claims were produced in experiments.
- I could not find the corresponding code. Hence, this work cannot be considered as demonstration for comprehensive codebase on interpreting multimodal networks with transformer architecture.

----- About the decision:

While the presented work is solid in its evaluation, it lacks either a empirical perspective on how generalizable the approach is to spacial or temporal concept changes, or a demonstration of the actionability of explanations, or a confirmation of the new data understanding provided, or an easy-to-apply codebase. Therefore, I estimate that the actual paper impact would be unsufficient, any of the previous being sufficient in my opinion for paper acceptation.

----- Points that I would like to see addressed but that did not impact the decision:

- Fig 1 and 11: Relative error color maps should be identically scaled to allow for model comparison.
- Weak representation due to point overlap in Fig 3 and 4 with lots of overlap. Providing a numerical aggregate (graphically or not) for the relation between |y1-y2| and cosine similarity would help, as visual inspection is not enough. For instance, spearman correlation.
- 330: AR being preferable due to low variability is a bit of a shortcut. It has been demonstrated in Fig 3 that the functional sensitivity of deeper layers is high, hence it it believable that a fidel explanation would also have high sensitivity. Provided reference Yeh and al do demonstrate that smoothed explanations for lower sensitivity reduce infidelity, but whether AR is a smoothed version of GA is not straighforward. Their general claim "if the explanation sensitivity is much larger than the function sensitivity around some input x, the infidelity measure in turn will necessarily be large for some point around x", does imply that explanation sensitivity much larger that model sensitivity impacts fidelity, but GA seems in line with the composition of layer 3 and 4 variability. To me, the conclusion I see is that GA tend to focus more on the more variable layers of the Transformer, while AR incorporate better stable layers.

**Questions:**

- 264: due to sampling, it is unclear if the training data from Transformer model training is used as test data for linear probing. This cannot be the case, to avoid evaluating overfit representations. Can this be confirmed?

---

> ### Author Response · Authors · 2024-11-19
> **Part 1/2**
>
> We thank the reviewer for their valuable feedback and constructive suggestions.
>
> **Weaknesses:**
>
> > Only one dataset is used, with no variation on modalities or indicators within each modalities. Even within the dataset, the authors do not truly enter a discussion on concept drift between the different fields, despite observing variability. Time-wise drift is not evaluated, despite years going from 2017 to 2023. How generalizable the approach is to other regions and future years is unknown.
>
> In addition to the page limitation, the current work prioritizes the interpretability analysis of the multimodal scenario, which is why we limited the extent of the modeling analysis. Nevertheless, multiple other studies have addressed the ablation analysis mentioned on a similar task, we modified the third paragraph in the Related Work section to mention some.
>
> > Expert understanding is missing from the evaluation, hence several of the explanations that the authors provide about data influence on the model cannot be verified. Sections after lines 371, 416, 460, and 501 lack this expert confirmation on data behaviour.
>
> While including expert knowledge in the analysis can provide insightful conclusions to validate the model behavior, it requires more information than we have, namely the farming practices adopted in each field and the phenological stages of the crop. Since we were able to obtain the latter information for certain soybean fields, we included the corresponding analysis in the rebuttal revision, as mentioned in the general comment.
>
> > Additionally, assertion line 419 could have been verified experimentally by computing attention distributions on 5 days averages, or 5 days min and max, instead of daily series. This would require fitting a new model on the modified data.
>
> Thank you for the suggestion, we conducted this analysis and included the corresponding analysis in Appendix E.2. It shows however that our assertion is only partially true. We adjusted the main text accordingly.
>
> > Despite actionability being mentionned in introduction, no such claims were produced in experiments.
>
> In the introduction we mention this as an ultimate goal of explaining multimodal networks, without claiming such insights in our contributions. Nevertheless, we attempt to provide an example of insightful understanding of the model behavior by comparing against established agronomical knowledge, and include the corresponding analysis, as described in the general comment.
>
> > I could not find the corresponding code. Hence, this work cannot be considered as demonstration for comprehensive codebase on interpreting multimodal networks with transformer architecture.
>
> We are working on preparing sharable code. Please refer to the last point in the general comment.
>
> ---
>
> **About the decision:**
>
> > While the presented work is solid in its evaluation, it lacks either a empirical perspective on how generalizable the approach is to spacial or temporal concept changes, or a demonstration of the actionability of explanations, or a confirmation of the new data understanding provided, or an easy-to-apply codebase. Therefore, I estimate that the actual paper impact would be unsufficient, any of the previous being sufficient in my opinion for paper acceptation.
>
> We are working on extending our work to other datasets, and have already included in the rebuttal revision examples of insightful interpretation of the results in the light of phenological stages and their known importance to the crop yield. We are also working on the code. Please refer to the general comments for more details.

---

> ### Author Response · Authors · 2024-11-19
> **Part 2/2**
>
> **Points that I would like to see addressed but that did not impact the decision:**
>
> > Fig 1 and 11: Relative error color maps should be identically scaled to allow for model comparison.
>
> We are working on reproducing the error maps for the final revision.
>
> > Weak representation due to point overlap in Fig 3 and 4 with lots of overlap. Providing a numerical aggregate (graphically or not) for the relation between |y1-y2| and cosine similarity would help, as visual inspection is not enough. For instance, spearman correlation.
>
> The suggested analysis and discussion is now added in Appendix E.1.
>
> > 330: AR being preferable due to low variability is a bit of a shortcut. It has been demonstrated in Fig 3 that the functional sensitivity of deeper layers is high, hence it it believable that a fidel explanation would also have high sensitivity. Provided reference Yeh and al do demonstrate that smoothed explanations for lower sensitivity reduce infidelity, but whether AR is a smoothed version of GA is not straightforward.
>
> We would not claim that AR is a smoothed version of GA, based on how smoothing is typically applied to feature attribution methods as described in Yeh et al.'s paper. However, it is worth noting that the non-smoothed use of gradients in GA raises concerns, as this approach is known to increase sensitivity and make predictions more susceptible to adversarial attacks, as highlighted in Yeh et al.'s work and originally demonstrated in [Ghorbani et al. 2019](https://ojs.aaai.org/index.php/AAAI/article/view/4252). We have further clarified this point in the related paragraph.
>
> > Their general claim "if the explanation sensitivity is much larger than the function sensitivity around some input x, the infidelity measure in turn will necessarily be large for some point around x", does imply that explanation sensitivity much larger that model sensitivity impacts fidelity, but GA seems in line with the composition of layer 3 and 4 variability.
>
> In fact, AR uses the raw attention matrices and better reflects their product (after adding an identity matrix to each one). GA, on the other hand, follows a similar process but multiplies each matrix with its gradient with respect to the output. Therefore, we doubt that a decision between AR and GA can be made solely based on the visual alignment between Figures 3 and 4, without considering their respective formulas.
>
> > To me, the conclusion I see is that GA tend to focus more on the more variable layers of the Transformer, while AR incorporate better stable layers.
>
> We understand your perspective but interpret it differently. Both methods incorporate the matrices from all layers: the chained multiplication of the attention distributions results in low-fluctuating weights (as indicated by AR), but GA introduces high fluctuations due to the inclusion of gradients in the chained multiplication. From another perspective, is the model sensitive? We don't believe so, as suggested by the prediction maps in Figures 1 and 11. Consequently, the feature attributions should similarly exhibit low sensitivity, which is why we preferred AR over GA.
>
> ---
>
> **Questions**
>
> > 264: due to sampling, it is unclear if the training data from Transformer model training is used as test data for linear probing. This cannot be the case, to avoid evaluating overfit representations. Can this be confirmed?
>
> The current data sampling approach does not prevent the model's training data from being used to test the linear probes. However, we do not see why this would be problematic. The goal of linear probes is to investigate the representations learned by the model, but could you clarify why a training sample would not be suitable for this evaluation?

---

> ### Comment · Reviewer_6E8s · 2024-11-21
>
> Thank you for the detailed answer. It is clear on what you could and could not do, and I appreciate the effort made on the phenological stages and the ablation study on appendix E.2, and other corrections.
>
> As for the comparison on AR and GA, it is precisely the gradients that leads me to think that later layers would be emphasized by GA, as $\frac{\delta A}{\delta y_t}$ with $y_t$ the class, is likely lower in earlier layers (for the same reasons as the vanishing gradient problem). Nevertheless, I lack experimental evidence, and preferring AR to GA based on sensitivity is understandable.
>
> On the last question, my concern was that representations of training samples might exhibit lower complexity and clearer structure due to the unavoidable overfitting that occurs during training. Test samples would better show how general the representations are.
>
> I understand that datasets cannot be shared, which is unfortunate as it prevents reproducibility. As mentioned in the global rebuttal, you will produce code that include models, interpretability methods and visualization functions, and I very much welcome this addition.
>
> Given the revision, clarifications and promise to release code, I will update my score.

---

> > ### Author Response · Authors · 2024-11-22
> >
> > While we were hoping for a higher increase in the rating, we sincerely appreciate your update and, above all, the helpful feedback you provided earlier.
> >
> > We will add a second general comment when we submit the final revision with the remaining changes.

---

### Official Review · Reviewer_GQRo · 2024-11-01

**Soundness:** 2
**Presentation:** 2
**Contribution:** 2
**Rating:** 3
**Confidence:** 4

**Summary:**

The paper introduces a framework based on Transformer models for crop yield prediction, emphasizing multimodal learning by integrating diverse data sources (e.g., satellite imagery, weather, and soil data) to improve prediction accuracy. A central contribution is the use of self-attention mechanisms within Transformers to achieve intrinsic interpretability, setting it apart from traditional post-hoc methods. The study finds that Transformer models not only enhance interpretability but also outperform other architectures in prediction accuracy.

**Strengths:**

- The paper introduces an original approach to crop yield prediction using Transformer-based models, a relatively novel application in this field. Its emphasis on intrinsic interpretability in multimodal learning stands out, offering a fresh alternative to the commonly used post-hoc methods in agricultural and environmental modeling.

- The methodology is robust, featuring extensive experimentation across multiple neural network architectures (e.g., LSTM, CNN, and Transformer) to identify the most effective model. Additionally, the study's comparative analysis of interpretability techniques—such as Attention Rollout, Generic Attention, and Shapley values—adds considerable depth, enhancing the understanding of the model’s interpretability from multiple perspectives.

- The paper is well-organized, with clear, accessible explanations of complex methodologies, complemented by visual aids that clarify the technical content and make intricate processes more intuitive for readers.

- The model’s capability to provide accurate, interpretable predictions at the sub-field level holds significant implications for agriculture, where trust in predictive models is crucial for effective decision-making.

**Weaknesses:**

- Although the study compares Transformer models with LSTM, ALSTM, and CNN architectures, including more contemporary multimodal learning approaches could further enhance the analysis. For instance, comparisons with models such as Multimodal Variational Autoencoders or Multimodal Contrastive Learning frameworks would provide a more comprehensive evaluation of the Transformer's effectiveness relative to recent innovations in multimodal integration.

- The Transformer model, with its 109,345 parameters, demonstrates an improvement in R² from 0.41 (ALSTM’s performance) to 0.46 and a reduction in MAE from 2.00 to 1.90. However, this comes with a significant increase in model complexity compared to ALSTM’s 38,017 parameters. A discussion on the computational costs, including inference time and resource requirements, would offer a balanced view of the model’s practicality, especially in resource-constrained settings.

- While the authors focus on a qualitative evaluation of interpretability, the assessment could be strengthened by incorporating functionally-grounded evaluation metrics. Adding objective measures such as fidelity and sparsity would provide a more comprehensive and robust assessment of the model's interpretability.

- Testing the model's adaptability across various crops and regions would help demonstrate its robustness. Expanding the dataset to incorporate diverse agricultural contexts or conducting transferability experiments across different geographic regions would offer insights into the model's broader applicability and generalization capacity.

**Questions:**

The primary issues were outlined in the weaknesses section, but there are additional questions and suggestions:

- Providing a more detailed description of the preprocessing steps for each data type would significantly enhance the reproducibility and transparency of the approach.

- Could you elaborate on how you determined the optimal balance between model complexity (e.g., number of Transformer layers and parameters) and interpretability? Insights into this decision process would clarify the rationale for the chosen model configuration.

- Attention to formatting and typographical consistency would improve readability. For example, there is a missing reference link in line 1122 ("same procedure described in Section ??"). A careful review for similar issues would enhance the paper’s presentation.

---

> ### Author Response · Authors · 2024-11-19
>
> We thank the reviewer for their valuable feedback.
>
> **W1: inclusion of models such as Multimodal Variational Autoencoders or Multimodal Contrastive Learning**
>
> While the suggested models are relevant to the task and have the potential to improve performance, we intentionally limited the modeling scope in this study to focus on the interpretability of *supervised learning* techniques.
>
> **W2 & Q2: computational costs and choice of Transformers over ALSTM**
>
> During training, both models required between 5 and 6 hours on a single GPU machine. As suggested, we included in the rebuttal revision a table in Appendix C summarizing the inference times for processing a batch of 1,000 samples on CPU and GPU machines. On a CPU machine, recurrent networks are very slower due to their sequential data processing, while Transformers are significantly faster. On a GPU machine, the parallel processing accelerates LSTM computations; however, all models maintain processing times mostly below 55 milliseconds. Therefore, we believe the Transformer model offers a well-balanced choice, considering CPU and GPU inference times, performance improvements, and interpretability.
>
> **W3: quantitative evaluation of the explanation methods**
>
> We recognize that quantitative evaluations would provide a more objective assessment of the methods we tested. Although we attempted to implement the sensitivity and infidelity metrics during this discussion phase, it appears that they will require more time to be properly adapted to the multimodal learning context. While we have already tested these metrics on unimodal networks, their implementation in this work is not straightforward.  We will include however this point as a future work in the conclusion section in the final revision.
>
> **W4: generalizability of the model to other crops and regions**
>
> The corresponding analysis will be included in the final revision. Please refer to the first point in the general comment.
>
> **Q1: preprocessing steps for each data type**
>
> We will include more detailed descriptions of the preprocessing steps per modality in the final revision.
>
> **Q3: attention to formatting and typographical consistency**
>
> Thank you for highlighting the typo. We have carefully reviewed the paper to avoid similar mistakes. Any additional comments on how to improve the presentation of the paper would be greatly appreciated.

---

### Official Review · Reviewer_NMdL · 2024-11-01

**Soundness:** 2
**Presentation:** 1
**Contribution:** 2
**Rating:** 3
**Confidence:** 3

**Summary:**

This paper addressed a crop yield prediction task in a multi-modal learning setting using multiple types of data such as satellite images and weather information.
In some prediction models, the paper showed that a transformer-based model achieved better predictive accuracy.
To understand the behaviors of the transformer model, it attempted to apply various existing explanation methods to the model.

**Strengths:**

- In the literature on crop yield prediction, efforts related to the explainability of this research may be novel.
- Comparative experiments and analyses have been conducted extensively.

**Weaknesses:**

- The importance of the crop yield prediction task is not adequately stated. Therefore, the usefulness and impact of the analysis results are not conveyed.
- The technical contributions of this research are unclear. Although the data may be unique, the prediction model and explanation methods used are existing techniques and appear to be merely applied to crop yield prediction.

**Questions:**

- Compared to other studies on explainability in multimodal learning, where do you believe the novelty lies?
- Are you predicting the yields of the three types of crops in a multi-task manner? For the results in Table 1, what crop yield values are considered the ground truth for evaluation?

---

> ### Author Response · Authors · 2024-11-19
>
> We thank the reviewer for their valuable feedback.
>
> **W1: importance of the crop yield prediction**
>
> That was indeed not clarified.  The yield prediction models can be used by digital farming services for early forecasting of the potential yield, and by insurance companies to compare reported and expected yield. Depending on the modalities used, it can also support farming practices to ensure and optimize the agricultural profitability. On larger scales, it can also support predicting the agricultural production on regional and national levels, and comparing it against current and future demand,  which also impacts market strategies to stabilize prices, plan export and import strategies, and ensure food security.  We modified the *Related Work* section to clarify the importance of yield prediction in agriculture and related sectors.
>
> **W2 & Q1: technical contributions and novely**
>
> To the best of our knowledge, the existing methods used (SVS, AR, and GA) have not been leveraged to explain multimodal learning, using self-attention-based Transformer networks. These methods are processed differently under this framework, and we additionally provide a new method to evaluate modality contributions, namely the ‘Weighted Modality Activations’, which can be considered as a variant of CAM adapted for multimodal networks. This combination of post-hoc and intrinsic techniques are compared, showing that the feature/modality attributions do not align across methods, and calling the community for more work addressing the interpretability of multimodal learning.
>
> **Q2: are the three crops predicted together?**
>
> we train a separate model for each crop. The main analysis focuses on corn, as stated in subsection 4.1, while the results for the wheat and soybean models have been included in the appendix. To improve clarity and avoid confusion, we have added references to the corn model in other parts of the text.
>
> **Q2: in Table 1, what crop yield values are considered the ground truth for evaluation?**
>
> if the question concerns the crop type, the focus is on corn. If the question is about how the yield ground-truth values are collected, we describe it in the same subsection: combine harvesters are equipped with yield estimators that measure yield in tons per hectare, calculated progressively as the combine drives through the field. The resulting yield maps are then rasterized to align with the spatial resolution of the satellite data (10m).

---

### Official Review · Reviewer_zjzB · 2024-11-03

**Soundness:** 2
**Presentation:** 3
**Contribution:** 1
**Rating:** 3
**Confidence:** 4

**Summary:**

This research paper explores intrinsic explainability techniques for multimodal learning, specifically focusing on crop yield prediction. The authors employ Transformer-based architectures to integrate diverse data sources, including satellite imagery, weather information, soil characteristics, and terrain data. The study evaluates various neural network designs, such as 1D-CNN, LSTM, ALSTM, and Transformer models. To interpret the model's decision-making process, the authors analyze learned representations across layers using linear probes, examine attention weight distributions within fields and across layers, and compare intrinsic attribution methods like Attention Rollout and Generic Attention with post-hoc techniques such as Shapley Values. The findings reveal that Transformer-based models not only excel in performance but also offer inherent interpretability benefits without sacrificing accuracy.

**Strengths:**

* This research paper presents a comprehensive analysis of a wide range of analytical approaches through a detailed ablation studies to compare various model architectures. Further, layer-wise analysis using linear probes provides insights into the evolution of learned representations across the model's depth.  Finally, exploration of attention weight distributions and comparison across multiple attribution methods is done.
* Evaluation on real-world data for three different crops results in direct applications in agriculture and remote sensing demonstrating that interpretability doesn't compromise performance.

**Weaknesses:**

* Choice of baseline models can be improved. For example, ConvLSTM [1], STATT [2], 3D-CNN [3] seems more suitable for satellite image time-series data.
* The focus on a dataset from Argentina raises questions about generalizability to other regions and crop types. Including experiments on datasets from various geographic regions would strengthen the model's applicability and reliability.
* It would be better if the interpretability results are accompanied with hypothesis/explanations derived from the domain knowledge (field/agronomic knowledge). For example, the modality impact scores does not match with agronomic significance, which makes it difficult to trust the internal mechanisms of the model.
* The temporal layer-wise attention are hard to interpret, where there doesn’t seem to be any correspondence with growing/harvesting season of the studies crops. The authors did a commendable job of dissecting the modern deep learning architectures, however the results does not seem to provide any meaningful insights.

References:

[1] Shi, Xingjian, Zhourong Chen, Hao Wang, Dit-Yan Yeung, Wai-Kin Wong, and Wang-chun Woo. "Convolutional LSTM network: A machine learning approach for precipitation nowcasting." Advances in neural information processing systems 28 (2015).

[2] Ghosh, Rahul, Praveen Ravirathinam, Xiaowei Jia, Chenxi Lin, Zhenong Jin, and Vipin Kumar. "Attention-augmented spatio-temporal segmentation for land cover mapping." In 2021 IEEE International Conference on Big Data (Big Data), pp. 1399-1408. IEEE, 2021.

[3] Ji, Shunping, Chi Zhang, Anjian Xu, Yun Shi, and Yulin Duan. "3D convolutional neural networks for crop classification with multi-temporal remote sensing images." Remote Sensing 10, no. 1 (2018): 75.

**Questions:**

1. Could you provide more validation of the interpretability results? For example, comparing the identified important temporal periods with known critical growth stages of crops?
2. The weighted modality activations and Shapley values show quite different results for modality importance. What's your hypothesis about these differences, and which method do you think is more reliable?
3. How do the interpretability results vary across different crops? Are there consistent patterns that align with known agricultural principles?

---

> ### Author Response · Authors · 2024-11-19
>
> We thank the reviewer for their valuable feedback and constructive suggestions.
>
> **W1: choice of baseline**
>
> The data is processed on a pixel-wise basis, where each 10x10m pixel is passed individually to the model. Thus, the satellite and weather data are treated as multivariate time series, while the soil and terrain elevation data are treated as static tabular data. We tried to cover the three main types of deep learning models commonly used for time-series/sequential data processing: recurrent, convolutional, and attention-based networks.
>
> **W2,3,4 and Q1:**
>
> We have incorporated the suggested analysis including domain knowledge in the interpretation in the rebuttal revision, and will address the extension of the study to other datasets in the final revision. For further details, please refer to the general comment.
>
> **Q2: modality importance between Shapley values and the Weighted Modality Activations**
>
>  Each method has its merits: SVS is grounded in game theory, while weighted modality activations are intrinsic to the model and are therefore expected to provide results more faithful to the model’s inner workings. This is mentioned in the final lines before the conclusion section. Further research is necessary to evaluate the correctness of both methods. We hope our work contributes to shedding light on this discrepancy and encourages additional research efforts to address the interpretability of multimodal networks.
>
> **Q3: results' variance across crops, and consistency with agricultural principles**
>
> Regarding temporal importance, we observed how the patterns might vary from a field to another (as seen in Figures 6 and 18 in the rebuttal revision). Collecting more information about the farming practices in each field is necessary to understand these differences. Additionally, as described in the general comment, for certain soybean fields we leverage the information about phenological stages to interpret the temporal attributions and verify their alignment with agricultural principle specific to this crop. We added the corresponding analysis to the appendix.
>
> Regarding the modality importance, some patterns are similarly observed across the three tested crops: the satellite modality is predominant according to the SVS results, while it contributes much less according to the weighted modality activations. We added more clarifications of the similarities in the text.

---

### Official Review · Reviewer_Y8Lw · 2024-11-08

**Soundness:** 2
**Presentation:** 2
**Contribution:** 1
**Rating:** 3
**Confidence:** 3

**Summary:**

This paper focuses on developing a model for crop yield prediction by exploring various model architectures and feature engineering to effectively create and merge feature representations from multiple modalities. The authors also experiment with different explainability techniques to analyze patterns between the model's internal representations and its predictions, specifically examining how these relate to the nature of each data modality.

**Strengths:**

The model development and analysis are thorough.

**Weaknesses:**

The paper lacks a clear novel contribution, as it primarily involves constructing a model for crop prediction and testing different model architectures and feature engineering methods. The authors identify a transformer-based architecture as the best-performing model and use various XAI methods, such as linear probes across layers and attention weights attribution, to investigate the relationship between internal representations and predictions based on data modality.

The experiments appear somewhat disjointed, with individual assumptions tested in isolation rather than in service of a coherent, high-level research question. Since the paper centres on explainability, it would be valuable to provide clear takeaways from these analyses, mainly to guide model builders, decision-makers, and practitioners in understanding the practical implications of the findings.

Overall, while the paper is sound, its contribution feels limited. It may be better suited for a more specialized venue focused on applied machine learning or exploratory studies rather than the conference's main track.

**Questions:**

1. Lines 46–47: You state, "Intrinsic explanations are inherently more faithful and less prone to errors introduced by surrogate models." I am not sure about the truthfulness of this claim.  Could you provide references that support it?

2. What is the rationale behind comparing the attention-based attribution and Shapley Values? Are you treating SV as a ground truth for attribution?

3. Table 1: Given the close performance metrics, including standard deviations for each metric would help determine if there are statistically significant differences between model performances.

---

> ### Author Response · Authors · 2024-11-19
>
> We regret that the reviewer is not convinced about the contribution of our work. While it is true that our study is applied to yield prediction,  the framework remains broadly applicable to multimodal learning networks using static and temporal tabular modalities for a regression task, and can be extended easily to classification tasks. As suggested by another reviewer, we are preparing to also publish the code to enable practitioners to use and evaluate our approach.
>
> Moreover, our evaluation of post-hoc against intrinsic explanation methods, and the observed discrepancy in some results, raises concerns about which method is more reliable to explain multimodal networks. While submitting such studies and results to specialized venue in agriculture or remote sensing is relevant, we chose to target more generalized machine learning venues to enhance visibility and engage the XAI community in addressing the interpretability challenges specific to multimodal networks.
>
>
> **Q1: "Intrinsic explanations are inherently more faithful and less prone to errors introduced by surrogate models."**
>
> The reference supporting this claim is cited in the previous sentence (Rudin 2019), while the references we cite afterward provide examples of post-hoc model-agnostic xai methods.  The claim is explicitly discussed in [Rudin’s paper](https://www.nature.com/articles/s42256-019-0048-x), on the second page, in the paragraph titled: "Explainable ML methods provide explanations that are not faithful to what the original model computes.”. Surrogate models are referred to as ‘approximations’.
>
> **Q2: the rationale behind comparing the attention-based attribution and Shapley Values**
>
> The main goal of comparing SVS with attention-based attributions is to evaluate how post-hoc methods perform relative to intrinsic techniques. We acknowledge the difficulty in defining ground-truth attributions, especially when using deep networks.
>
> **Q3: standard deviation over the performance metrics**
>
> Unfortunately, we doubt that we will be able to conduct the necessary experiments in time to include these results..

---

### Author Response · Authors · 2024-11-19
**General comments**

**Extension to other regions:**

As mentioned by reviewers [zjzB](https://openreview.net/forum?id=dQpZolwXiH&noteId=sUsLeiiCh9), [GQRo](https://openreview.net/forum?id=dQpZolwXiH&noteId=j6tJTfeh7j), and [6E8s](https://openreview.net/forum?id=dQpZolwXiH&noteId=1OaJ7ZVgbW), limiting our analysis to data from Argentina might not reflect the generalizability of our approach to other regions. Such extensions are typically challenging in this field, as precise yield data maps are often scarce and expensive to obtain. Yet, we are currently working on including analysis of other regions, and will post an update as soon as we get the corresponding results.

**Agronomical interpretation & verification of the results:**

Reviewers [zjzB](https://openreview.net/forum?id=dQpZolwXiH&noteId=sUsLeiiCh9) and [6E8s](https://openreview.net/forum?id=dQpZolwXiH&noteId=1OaJ7ZVgbW) raised concerns about the utility of the results within the agricultural context, in order to derive insightful conclusions. To address this concern, a key information that is not easy to obtain is the phenological stages of the crop. While such information was not collected along with the ground-truth yield, we were able to obtain approximations of the growth stages for certain soybean fields. In the rebuttal revision, we included the analysis in Appendix F, leveraging this information to interpret the temporal attribution of satellite and weather modalities accordingly, and to verify the patterns learned by the model against established knowledge from the agricultural field.

**Code:**

As strongly recommended by reviewer [6E8s](https://openreview.net/forum?id=dQpZolwXiH&noteId=1OaJ7ZVgbW), we are in the process of preparing a repository to share the code for our approach. Although the dataset cannot be published at this time, we will make available sufficient parts of the modeling and interpretation analysis to enable the use and testing of our framework. We hope we can timely share an initial draft during the discussion phase.

---

### Note · Authors · 2024-11-25

**Comment:**

We sincerely thank the reviewers for their thoughtful comments and the time and effort to review our manuscript.

Due to an error identified in our code, we suspect the current results may not be entirely reliable, and thus we have decided to withdraw our submission in order to rerun the complete analysis.

**Withdrawal Confirmation:**

I have read and agree with the venue's withdrawal policy on behalf of myself and my co-authors.